# From Volume to Value:
# Preference-Aligned Memory Construction for On-Device RAG

Changmin Lee [1]   Jaemin Kim [1]   Taesik Gong [1]

## Abstract

With the rapid emergence of personal AI agents based on Large Language Models (LLMs), implementing them on-device has become essential for privacy and responsiveness. To handle the inherently personal and context-dependent nature of real-world requests, such agents must ground their generation in device-resident personal context. However, under tight memory budgets, the core bottleneck is *what to store* so that retrieval remains aligned with the user. We propose EPIC (Efficient Preference-aligned Index Construction), which focuses on user preferences as a compact and stable form of personal context and integrates them throughout the RAG pipeline. EPIC selectively retains preference-relevant information from raw data and aligns retrieval toward preference-aligned contexts. Across four benchmarks covering conversations, debates, explanations, and recommendations, EPIC reduces indexing memory by 2,404×, improves preference-following accuracy by 18.79%p, and achieves 32.17× lower retrieval latency over the best-performing baseline. In on-device experiments, EPIC maintains under 1 MB memory and achieves 5.21 to 29.35 ms/query latency across three platforms, while supporting streaming updates under preference drift. Our code and data are available at https://github.com/UbiquitousAILab/EPIC.

## 1. Introduction

Recent advances in Large Language Models (LLMs) have accelerated the shift toward personal AI agents that assist with daily tasks (Park et al., 2023; Xi et al., 2025;

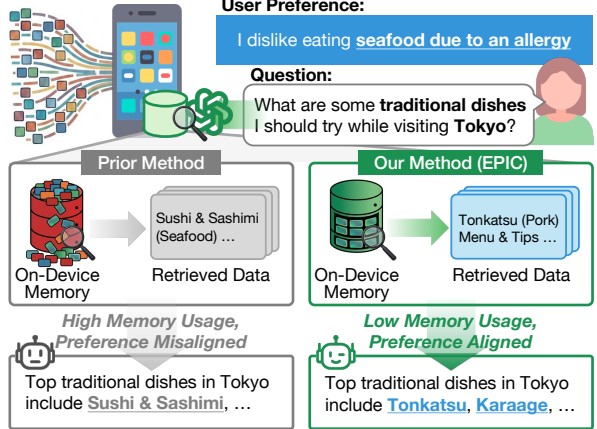

*Figure 1.* Prior Method indiscriminately stores raw data, which is infeasible under tight on-device memory budgets and can yield preference-misaligned responses (left). EPIC instead retains only preference-relevant data with aligned instructions, enabling efficient retrieval and preference-aligned responses (right). Example from the PrefWiki dataset.

Li et al., 2025; 2024b; de Barcelos Silva et al., 2020; Lee et al., 2024; Li et al., 2024a). Prior studies on assistant usage and mobile search behavior suggest that a substantial fraction of real-world requests are inherently personal and context-dependent (Jiang et al., 2015; Guy, 2016). This makes personalization a core requirement for practical assistants: users expect assistants to remember their tastes, choices, and constraints (Zhao et al., 2025). The effectiveness of such personalization depends on how well an agent can ground user intent in device-resident contexts.

To this end, the agent should leverage on-device interaction footprints to condition responses on personal context. However, personal devices generate a diverse range of heterogeneous data, ranging from large-scale static document corpora (e.g., Wikipedia) to dynamic digital footprints such as web pages visited during browsing, frequently updated news, social media feeds, and conversation histories. Since such data may contain sensitive information, storing it on external servers is often unacceptable due to privacy constraints (Sweeney, 2000; Abadi et al., 2016; Shokri & Shmatikov, 2015; Neverova et al., 2016). Therefore, such data must be stored on-device; yet storing it in its entirety is infeasible under strict memory constraints (Park et al.,

---

[1]Department of Computer Science and Engineering, Ulsan National Institute of Science and Engineering, Ulsan, Republic of Korea. Correspondence to: Taesik Gong <taesik.gong@unist.ac.kr>.

*Proceedings of the 43rd International Conference on Machine Learning*, Seoul, South Korea. PMLR 306, 2026. Copyright 2026 by the author(s).

2025), being a critical bottleneck for personalized agents.

Meanwhile, Retrieval-Augmented Generation (RAG) offers a promising solution to ground LLMs in external knowledge without retraining (Lewis et al., 2020; Lee et al., 2019; Karpukhin et al., 2020; Ram et al., 2023). Prior work personalizes RAG by constructing retrieval memory from user-specific artifacts (Wang et al., 2024; Mysore et al., 2024) or by rewriting queries before retrieval (Zhou et al., 2024; Zhang et al., 2026). However, under strict on-device privacy and memory budgets, the central bottleneck shifts from *how to use* personal information at retrieval time to *what to store* in the first place. We therefore argue for a fundamental shift toward *efficient memory construction*, specifically determining:

> ***What should be stored for on-device personalized memory under tight budgets?***

Instead of indiscriminately storing all incoming data, we propose a novel approach that constructs on-device memory by selectively storing only the information relevant to personal contexts.

Among various forms of personal context, *user preferences* provide a compact and stable abstraction of what consistently matters across interactions (Purificato et al., 2024; Wei et al., 2025). Unlike transient contextual signals, preferences tend to persist over time and play a central role in user satisfaction (Kiseleva et al., 2016), and can be inferred from device-generated data (e.g., conversational histories) with LLMs even when implicitly expressed (Kim et al., 2025; Wang et al., 2025). As depicted in Figure 1, our approach filters the raw data to retain only preference-related data, significantly reducing memory use while ensuring the retrieved documents match the user's preference. This motivates *preference-aligned memory construction*: under tight on-device budgets, the system should prioritize storing information that is relevant to a user's preferences and constraints.

In this paper, we propose EPIC (**E**fficient **P**reference-aligned **I**ndex **C**onstruction), a framework for building compact, preference-aligned on-device memory from raw data under privacy constraints. EPIC consists of three core components.

First, *Semantic-Based Coarse Filtering* rapidly performs high-recall pruning of preference-irrelevant content by leveraging proximity within the latent embedding space. Next, *Preference-Aligned Fine Verification* grounds latent relevance in explicit semantics by leveraging a language model to perform fine-grained verification for selected data and generate anchor instructions that explain content-preference alignment. Finally, *Preference-Guided Query*

*Steering* transforms the query representation by shifting it toward the selected preference direction in embedding space, enabling more precise preference-aligned retrieval with negligible computational overhead. Together, these components enable compact on-device memory that remains preference-faithful under strict constraints.

Evaluating EPIC requires assessing preference-aligned responses in settings where the ground-truth answer is often open-ended. To this end, we adopt the rigorous evaluation metrics and dataset construction protocol of PrefEval (Zhao et al., 2025), a preference-centric benchmark originally developed for conversation histories. To reflect a wider range of real-world scenarios beyond conversational memory, we introduce three new benchmarks, **PrefWiki**, **PrefRQ**, and **PrefELI5**, covering static knowledge corpora and web-derived digital footprints.

Our results highlight that EPIC is highly efficient, reducing stored memory by **2,404×** while improving personalization accuracy by **18.79%p** and shortening retrieval latency by **32.17×** compared to the best-performing baseline (Lee et al., 2025). To validate practical deployability, we conducted on-device experiments across three edge and consumer-level platforms: a Jetson Orin Nano, a MacBook Pro M4, and a Galaxy Z Flip 6. EPIC maintains a compact memory footprint under **1 MB** and achieves **5.21** to **29.35 ms** retrieval latency across these devices, with only marginal query-steering overhead.

## 2. Related Work

### 2.1. Retrieval and Indexing for RAG

RAG has been widely studied as a framework for grounding LLM outputs in relatively static external corpora, such as large document collections or curated knowledge bases (Lee et al., 2019; Lewis et al., 2020; Gao et al., 2023). Retrieval has advanced from sparse methods such as BM25 (Robertson et al., 1995) to dense retrievers such as DPR (Karpukhin et al., 2020) and Contriever (Izacard et al., 2022), and recently to large-scale embedding models such as NV-Embed (Lee et al., 2025). Beyond retrievers, recent systems structure corpora into hierarchical or graph-based representations to improve retrieval and downstream answering: RAPTOR (Sarthi et al., 2024) builds hierarchical summaries, while HippoRAG (Gutiérrez et al., 2024) and HippoRAG 2 (Gutiérrez et al., 2025) propagate relevance over document-entity graphs. However, these approaches largely assume static or centrally managed corpora and focus on improving retrieval quality given a fixed index. In on-device settings with ever-growing device-resident data, indiscriminate indexing becomes the bottleneck, motivating *selective memory construction* that decides *what to store* for resource-efficient, preference-aligned memory.

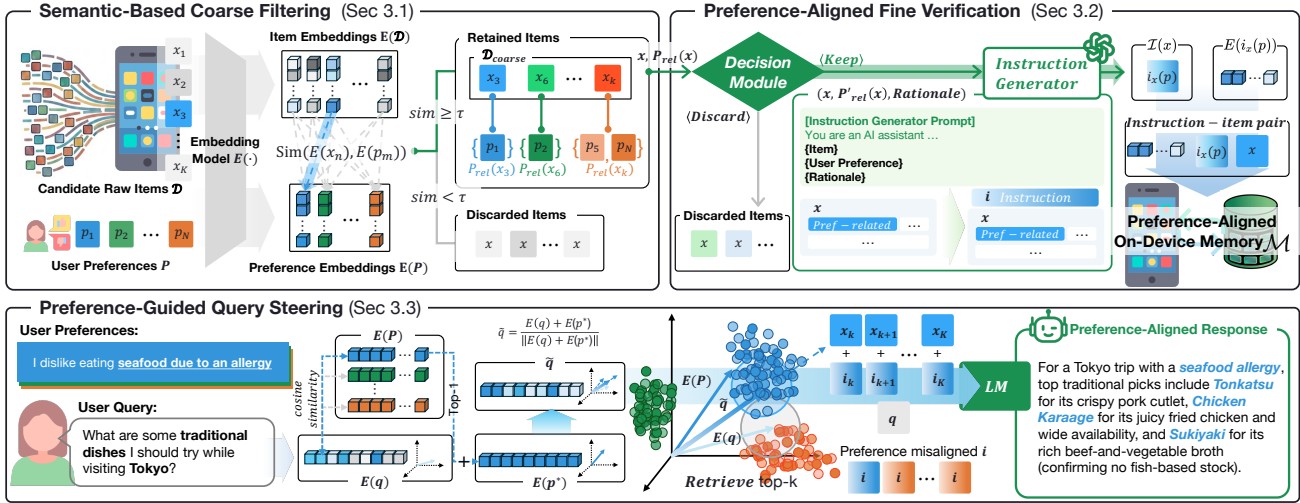

*Figure 2.* **Overview of EPIC's pipeline.** (i) *Semantic-Based Coarse Filtering* (Sec. 3.1): documents from a large corpus are first encoded and compared with user preference embeddings; only those with at least one preference-aligned match pass this stage. (ii) *Preference-Aligned Fine Verification* (Sec. 3.2): the *Decision Module* verifies textual alignment and discards unrelated documents, while the *Instruction Generator* synthesizes preference-conditioned anchor instructions for the retained items. (iii) *Preference-Guided Query Steering* (Sec. 3.3): user-query embeddings are steered toward their associated preference directions, enabling the language model to produce preference-aligned responses.

## 2.2. Personalization in RAG

Recent lines of work personalize RAG by incorporating personal signals beyond query relevance. One direction personalizes on the *memory side* by constructing retrieval memory from user-specific artifacts, such as user-authored documents, profiles, or curated interaction histories. Representative examples include EMG-RAG (Wang et al., 2024), which organizes device-derived memories into an editable graph to support downstream retrieval and interaction; and PEARL (Mysore et al., 2024), which selects user-authored content to capture individual style and values. Another direction injects user information on the *query side* by rewriting or expanding the input query before retrieval. For instance, Cognitive Personalized Search (Zhou et al., 2024) injects user context via query rewriting, and Personalize Before Retrieve (PBR) (Zhang et al., 2026) studies LLM-based personalized query expansion prior to retrieval.

While effective, most prior work starts from an existing set of user artifacts or memories and focuses on *how* personal signals guide retrieval at query time. In contrast, on-device agents face raw, heterogeneous data under tight privacy and storage budgets; indiscriminate storage is infeasible. We therefore shift the focus to deciding *what to store* while jointly enhancing indexing and retrieval with minimal computational overhead, so that the resulting on-device memory remains both compact and preference-aligned.

## 3. Method

**Problem setting and challenges.** We study preference-aware Retrieval-Augmented Generation (RAG) for on-device personal agents. These agents must build a compact memory from heterogeneous and ever-growing information sources, ranging from static knowledge corpora to interaction-derived digital footprints and conversation histories collected during daily use. Unlike conventional RAG benchmarks with static curated corpora, low-relevance data on personal devices wastes precious memory and hinders the retrieval process from accurately reflecting actual user intents. This makes indiscriminate indexing both resource-prohibitive and prone to preference misalignment. Addressing these issues involves two fundamental challenges. First, *memory scalability* becomes a critical bottleneck: as information accumulates, storing everything quickly exhausts available storage on personal devices, requiring selective retention of data likely to be useful in future interactions. Second, *preference misalignment* arises during retrieval: standard retrievers optimize query-text similarity and are largely preference-agnostic (Zhao et al., 2025). As a result, retrieved content may be factually correct but inconsistent with the user's preferences, which can reduce the usefulness of generated responses. Accordingly, our objective is not to infer user preferences from raw logs, but to construct and retrieve compact, preference-aligned memory given a preference set. We assume throughout this work that such a preference set is available at indexing time, either explicitly provided by the user or obtained from a separate preference extraction pipeline.

**Method overview.** To address these challenges, we propose *EPIC* (**Efficient Preference-aligned Index Construction**), a framework for building compact, preference-aligned on-device memory for RAG (Figure 2). EPIC treats personalization as a memory construction problem: it decides *what to store* on-device so that retrieval is both query-relevant and preference-aligned. Concretely, EPIC integrates user preferences into both memory construction and retrieval via three components. The process begins with (i) **Semantic-Based Coarse Filtering** (Section 3.1), which efficiently discards the majority of preference-irrelevant content by exploiting geometric proximity within the latent embedding space to achieve high-recall pruning. This is followed by (ii) **Preference-Aligned Fine Verification** (Section 3.2), a stage that bridges the gap between approximate latent similarity and explicit semantic relevance by employing a language model to strictly validate alignment and generate explanatory anchor instructions. Finally, EPIC employs (iii) **Preference-Guided Query Steering** (Section 3.3) to dynamically modulate the query representation, shifting it toward the target preference direction in the embedding space to ensure preference-aligned retrieval with marginal computational overhead. Together, these components enable grounded and preference-consistent responses under strict on-device constraints.

## 3.1. Semantic-Based Coarse Filtering

In this stage, we embed data items and preferences into a shared semantic space to leverage the vector alignment between their representations. The computation is efficient, involving only embedding generation and similarity scoring, without requiring additional model-based inference or reasoning. This stage serves as an initial filter that rapidly discards clearly irrelevant items while retaining potentially useful ones for further verification.

Specifically, let $\mathcal{D}$ denote the set of candidate items encountered on-device (e.g., passages from static corpora or interaction-derived traces), and $P = \{p_1, p_2, \ldots, p_N\}$ denote the available user preference set. We embed each item $x \in \mathcal{D}$ and preference $p \in P$ using a shared sentence encoder $\text{Enc}_\theta$ (e.g., Contriever), obtaining a sentence embedding $g_\theta(t) \in \mathbb{R}^d$ for input text $t$ (either an item $x$ or a preference $p$). We denote the $\ell_2$-normalized embedding of $t$ as $\text{E}(t) = g_\theta(t)/\|g_\theta(t)\|_2 \in \mathbb{R}^d$.

Using these embeddings, we identify which preferences are semantically related to an item by computing cosine similarity $\text{Sim}(\cdot, \cdot)$ and selecting those above a threshold $\tau$:

$$P_{\text{rel}}(x) = \{p \in P \mid \text{Sim}\big(\text{E}(x), \text{E}(p)\big) \geq \tau\}, \quad \forall x \in \mathcal{D}. \tag{1}$$

$P_{\text{rel}}(x)$ denotes the subset of preferences semantically aligned with an item $x$. We then retain the candidate set

$\mathcal{D}_{\text{coarse}}$ (Eq. 2), containing items matched to at least one preference.

$$\mathcal{D}_{\text{coarse}} = \big\{x \in \mathcal{D} \mid P_{\text{rel}}(x) \neq \emptyset\big\}. \tag{2}$$

For each retained item $x \in \mathcal{D}_{\text{coarse}}$, both $x$ and its associated preference set $P_{\text{rel}}(x)$ are passed to the subsequent fine-grained verification stage.

## 3.2. Preference-Aligned Fine Verification

While coarse filtering efficiently prunes the candidate set, embedding-level similarity alone cannot capture nuanced text-level alignment. Our second stage therefore verifies preference alignment and constructs preference-aware memory entries in the form of explicit usage instructions. To this end, we introduce two complementary components that leverage the language understanding capabilities of an LM: a Decision Module (*DM*), which determines whether each candidate item should be discarded or retained for memory construction, and an *Instruction Generator* (*IG*), which generates preference-conditioned usage instructions associated with each retained item. The *DM* ensures that only items with genuine preference relevance are retained, while the *IG* produces concise directives that specify *how* and *when* the retained content should be used for preference-aligned response generation.

Together, these modules enable instruction-centric memory construction, so that downstream retrieval operates over compact instruction embeddings and retrieves the linked items for grounded, preference-consistent generation.

### 3.2.1. DECISION MODULE

For each candidate item $x \in \mathcal{D}_{\text{coarse}}$ and its associated preference set $P_{\text{rel}}(x)$, the *DM* takes both the item and the preferences as input and returns a structured output following a fixed schema (Appendix E):

$$DM(x, P_{\text{rel}}(x)) = (\text{Decision}, \text{Rationale}, P'_{\text{rel}}(x)). \tag{3}$$

where $\text{Decision} \in \{\langle\text{Keep}\rangle, \langle\text{Discard}\rangle\}$ indicates retention, $\text{Rationale}$ justifies the decision, and $P'_{\text{rel}}(x) \subseteq P_{\text{rel}}(x)$ is the refined preference subset directly relevant to $x$. The verified candidate set is then defined as:

$$\mathcal{D}_{\text{fine}} = \big\{x \in \mathcal{D}_{\text{coarse}} \mid DM\big(x, P_{\text{rel}}(x)\big)_{\text{Decision}} = \langle\text{Keep}\rangle\big\}. \tag{4}$$

If the decision for an item $x$ is $\langle\text{Discard}\rangle$, it is removed from further consideration, and no memory entry is created.

### 3.2.2. INSTRUCTION GENERATOR

If the decision is $\langle\text{Keep}\rangle$, the item $x$, its final preference set $P'_{\text{rel}}(x)$ and the $\text{Rationale}$ are passed to the *Instruction Generator*, which produces one or more preference-aware instructions:

$$\mathcal{I}(x) = \{ i_x(p) \mid p \in P'_{\text{rel}}(x) \}, \quad \forall x \in \mathcal{D}_{\text{fine}} \quad (5)$$

where $i_x(p) \coloneqq IG(x, p, \text{Rationale})$ is the instruction generated for item $x$ based on preference $p$. Each instruction is a concise, preference-conditioned directive that specifies *how* and *when* the item should be used for response generation under the given preferences (Appendix E for the prompt and Appendix F for examples).

### 3.2.3. INSTRUCTION-CENTRIC MEMORY

For each instruction $i_x(p) \in \mathcal{I}(x)$, we form an *instruction-item pair* $(i_x(p), x)$ and embed the instruction as $\mathrm{E}(i_x(p))$. The on-device memory is indexed by instruction embeddings, and each memory entry stores the corresponding item together with its instruction.

This instruction-centric memory construction captures personalization through explicit usage instructions while preserving the original content of retained items. The final memory $\mathcal{M}$ stores instruction–item pairs with their metadata:

$$\mathcal{M} = \left\{ \Big( x, \ i_x(p), \ p, \ \mathrm{E}\big(i_x(p)\big) \Big) \right.$$
$$\left. \Big| \ x \in \mathcal{D}_{\text{fine}}, p \in P'_{\text{rel}}(x) \right\}, \quad (6)$$

where $\mathcal{M}$ denotes the final instruction-centric memory, with each entry containing the raw item $x$, the specific instruction $i_x(p)$ generated for preference $p$, the corresponding preference itself $p$, and the instruction embedding $\mathrm{E}\big(i_x(p)\big)$ for all $x \in \mathcal{D}_{\text{fine}}$. By indexing instructions rather than raw items, the memory explicitly encodes preference-aware usage at the text level while remaining compact. During retrieval, matched instructions guide context selection toward preference-relevant items, while the linked items provide the factual grounding for generation.

### 3.3. Preference-Guided Query Steering

In EPIC, instruction embeddings are constructed to be preference-aware, but a raw query embedding may not sufficiently reflect the user's preference space. To bridge this gap, we introduce *preference-guided query steering*, which steers the query embedding toward the most relevant preference's direction to retrieve preference-aligned instructions.

Given a user query $q$, we select the most similar preference

$$p^* = \arg\max_{p \in P} \mathrm{Sim}\big(\mathrm{E}(q), \mathrm{E}(p)\big), \quad (7)$$

and form the steered query:

$$\tilde{q} = \frac{\mathrm{E}(q) + \mathrm{E}(p^*)}{\big\|\mathrm{E}(q) + \mathrm{E}(p^*)\big\|_2}. \quad (8)$$

*Table 1.* **Summary of our constructed preference-aware RAG benchmarks.** The columns # Doc, # Per, # Pref, and # Q denote the number of documents, personas, preferences, and questions, respectively. Representative examples of preference-question pairs are provided in Appendix D.4.

| Dataset | Task | Corpus (# Doc) | # Per | # Pref | # Q |
|---|---|---|---|---|---|
| **PrefWiki** | Recommendation | Wikipedia (6.9M) | 57 | 570 | 2,850 |
| **PrefRQ** | Debate | Wikipedia (6.9M) | 90 | 900 | 900 |
| **PrefELI5** | Explanation | Common Crawl (16.4M) | 73 | 730 | 730 |
| **PrefEval** | Conversation | LMSYS-Chat (1M) | 57 | 570 | 570 |

We implement retrieval with a FAISS (Facebook AI Similarity Search) index over instruction embeddings (Johnson et al., 2019). At query time, we perform nearest-neighbor search using $\tilde{q}$ to retrieve top-$k$ instructions, each pointing to its linked item for augmentation.

Overall, query steering improves the compatibility between user queries and preference-conditioned instruction embeddings, enabling preference-aligned instruction retrieval with factual grounding from the linked items.

## 4. Benchmarks for Preference-Aligned Memory Construction

**Evaluation Challenge.** Evaluating EPIC requires assessing whether generated responses remain preference-aligned even when the ground-truth answer is often open-ended. We therefore adopt the rigorous preference-centric evaluation protocol of PrefEval (Zhao et al., 2025), which provides LLM-based metrics and a dataset construction procedure grounded in explicit user preferences.

**Benchmark construction.** PrefEval primarily targets conversational memory and does not cover the broader range of heterogeneous data sources encountered by on-device personal agents. To reflect realistic on-device scenarios, we build a comprehensive benchmark suite spanning three data domains: (i) *static knowledge-based corpora*, (ii) *noisy web-derived digital footprints*, and (iii) *conversation histories*. Building on PrefEval, we construct **PrefWiki** and **PrefRQ** for static corpora, **PrefELI5** for web-derived footprints, and use **PrefEval** for conversation histories. These datasets serve as controlled proxies for on-device preference memory: they retain the validated preference-following evaluation protocol of PrefEval while varying the information sources that a personal device may encounter. Across all benchmarks, preference-question pairs are generated and validated using adapted prompts from PrefEval (Zhao et al., 2025). This procedure ensures that preferences meaningfully align with or conflict with the associated requests. Table 1 summarizes the benchmarks, and Appendix B and Appendix D provide sampling and dataset construction details.

*Table 2.* **Overall results.** We report preference-following accuracy (%) across three LLM backends. All methods are evaluated under the same PrefEval generation-and-judge protocol, using LLaMA-3.3-70B-Instruct as the judge. Best results are in bold.

| | Method | Qwen3-4B-Instruct-2507 | | | | Llama-3.1-8B-Instruct | | | | gpt-oss-20b | | | |
|---|---|---|---|---|---|---|---|---|---|---|---|---|---|
| | | PrefWiki | PrefRQ | PrefELI5 | PrefEval | PrefWiki | PrefRQ | PrefELI5 | PrefEval | PrefWiki | PrefRQ | PrefELI5 | PrefEval |
| | *Standard RAG* | | | | | | | | | | | | |
| | BM25 (Robertson et al., 1995) | 18.74 | 45.44 | 73.84 | 30.92 | 38.56 | 64.22 | 69.04 | 27.97 | 40.84 | 87.33 | 73.90 | 26.98 |
| | Contriever (Izacard et al., 2022) | 28.56 | 65.00 | 78.90 | 33.51 | 40.88 | 68.11 | 69.45 | 27.89 | 41.16 | 87.22 | 75.49 | 29.84 |
| | NV-Embed-v2 (Lee et al., 2025) | 37.82 | 65.22 | 79.45 | 32.98 | 44.53 | 70.22 | 69.86 | 30.88 | 42.29 | 86.56 | 77.01 | 29.28 |
| | *Indexing-enhanced RAG* | | | | | | | | | | | | |
| | RAPTOR (Sarthi et al., 2024) | 34.11 | 60.89 | 78.77 | 38.07 | 40.70 | 66.67 | 65.48 | 31.05 | 41.86 | 86.33 | 75.48 | 28.07 |
| | HippoRAG (Gutiérrez et al., 2024) | 24.53 | 19.22 | 74.25 | 20.70 | 38.42 | 54.00 | 71.37 | 21.40 | 39.47 | 89.86 | 76.24 | 26.37 |
| | HippoRAG 2 (Gutiérrez et al., 2025) | 36.53 | 68.11 | 80.55 | 33.51 | 42.91 | 66.00 | 69.73 | 32.98 | 39.58 | 86.56 | 75.21 | 27.19 |
| | *Preference-conditioned RAG* | | | | | | | | | | | | |
| | Pref-QR (Zhou et al., 2024) | 22.00 | 59.44 | 78.63 | 32.63 | 39.72 | 71.33 | 69.59 | 34.21 | 42.23 | 89.00 | 74.41 | 29.47 |
| | PBR (Zhang et al., 2026) | 28.42 | 64.33 | 79.04 | 32.98 | 41.02 | 70.67 | 70.14 | 30.70 | 41.19 | 87.00 | 75.07 | 27.54 |
| | **EPIC** | **44.95** | **69.78** | **87.95** | **61.93** | **54.07** | **83.00** | **87.95** | **65.61** | **73.26** | **93.89** | **87.61** | **77.96** |

*(Row label rotated on left: Accuracy (%))*

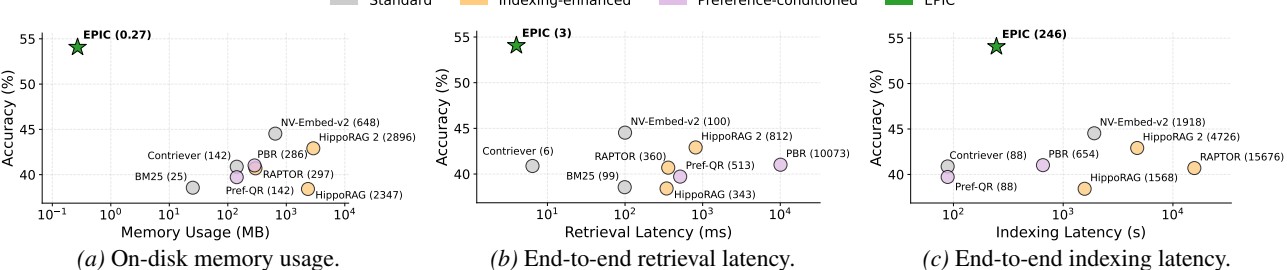

*(a)* On-disk memory usage.     *(b)* End-to-end retrieval latency.     *(c)* End-to-end indexing latency.

*Figure 3.* **Efficiency comparison across baselines.** We report on-disk memory usage, end-to-end retrieval latency, and indexing latency (detailed results in Appendix B.5). Numbers in parentheses represent the specific values on the x-axis for each method.

## 5. Experiments

### 5.1. Setup

**Baselines.** To evaluate the effectiveness of EPIC in constructing compact, preference-aligned memory for on-device agents from raw data, we compare against three categories of RAG baselines: *Standard RAG*, *Indexing-enhanced RAG*, and *Preference-conditioned RAG* (query-side preference conditioning). First, we consider a standard RAG pipeline that indiscriminately indexes all available items and retrieves the top-$k$ most similar items using FAISS (Johnson et al., 2019). We test three representative retrievers: the classic sparse matcher **BM25** (Robertson et al., 1995), the dense dual-encoder **Contriever** (Izacard et al., 2022), and the large-scale embedding model **NV-Embed** (Lee et al., 2025). Next, we compare against recent indexing-enhanced RAG frameworks that improve retrieval via structured organization beyond simple vector similarity: **RAPTOR** (Sarthi et al., 2024), which performs hierarchical clustering and summarization; **HippoRAG** (Gutiérrez et al., 2024); and **HippoRAG 2** (Gutiérrez et al., 2025), both of which exploit document-entity graphs to propagate relevance and support multi-document reasoning. For all baselines except BM25 and NV-Embed-v2, we use Contriever as the underlying retriever. Finally, we include two preference-conditioned RAG frameworks: **Pref-QR** (Preference-conditioned Query Rewriting), which rewrites the query by conditioning on explicit user preferences using the prompt template from Cognitive Personalized Search (Zhou et al., 2024), and **PBR** (Zhang et al., 2026), which generates preference-conditioned query expansions before retrieval.

**Comparison settings.** Overall, existing baselines either improve retrieval via structured indexing or condition the query on user preferences, but none directly optimize what to store as a preference-aligned on-device memory under tight resource budgets. For fair comparison across all baselines, we report the main results under a unified server setting with an identical LLM and embedding model.

**Evaluation protocol and metrics.** Our evaluation focuses on whether the constructed on-device memory enables preference-aligned retrieval. To isolate the effect of memory construction and retrieval, we adopt the PrefEval answer-generation and LLM-as-a-judge evaluation protocol (Zhao et al., 2025) across all methods, without modification (Appendix A.5 for details). We fix prompts, decoding parameters, and the answer generator so that performance differences arise solely from the retrieved content.

*Table 3.* **On-device latency breakdown across platforms.** Retrieval latency is measured per query, and indexing latency is measured per incoming item.

| Component | Jetson Orin Nano | MacBook Pro M4 | Galaxy Z Flip 6 |
|---|---|---|---|
| *Retrieval Latency (ms)* | | | |
| Embedding query | 29.03 | 3.36 | 3.85 |
| Preference-Guided Query Steering (Sec. 3.3) | 0.18 | 0.03 | 0.55 |
| FAISS retrieval | 0.14 | 1.84 | 0.81 |
| **Total Retrieval Latency** | **29.35** | **5.23** | **5.21** |
| *Indexing Latency (ms)* | | | |
| Embedding items | 29.99 | 4.31 | 4.01 |
| Semantic-Based Coarse Filtering (Sec. 3.1) | 0.02 | 0.01 | 0.10 |
| Preference-Aligned Fine Verification (Sec. 3.2) | 70.73 | 47.09 | 245.65 |
| Embedding instruction | 0.02 | 0.01 | 0.00 |
| Build FAISS | 0.00 | 0.00 | 0.00 |
| **Total Indexing Latency** | **102.67** | **51.42** | **249.76** |

Each response is evaluated using PrefEval's structured rubric, which assigns four binary error labels: preference-unaware, preference hallucination, inconsistency, and unhelpfulness. We report *preference-following accuracy* as the ratio of responses with no such errors. For evaluation, we use LLaMA 3.3 70B-Instruct (Grattafiori et al., 2024).

Beyond accuracy, we report three efficiency metrics that are critical for on-device deployment: *Memory Usage*, *Retrieval Latency*, and *Indexing Latency*. Memory Usage measures the on-disk footprint of each method's full retrieval state (e.g., stored items and auxiliary structures); Retrieval Latency measures end-to-end retrieval time per query; and Indexing Latency measures the total time to construct the retrieval state. Because methods maintain different components (e.g., vector indexes, summaries, graphs, or metadata), we provide the exact measurement protocol and inclusion rules in Appendix B.5.

### 5.2. Performance Analysis

Table 2 reports preference-following accuracy on four benchmarks (PrefWiki, PrefRQ, PrefELI5, and PrefEval) with three LLM backends: Qwen3-4B-Instruct-2507 (Qwen Team, 2025), Llama-3.1-8B-Instruct (Grattafiori et al., 2024), and gpt-oss-20b (OpenAI, 2025). Across all datasets and backends, EPIC consistently achieves the highest accuracy among standard RAG baselines (BM25, Contriever, NV-Embed-v2), indexing-enhanced RAG frameworks (RAPTOR, HippoRAG, HippoRAG 2), and preference-conditioned RAG baselines (Pref-QR, PBR). For instance, under the Llama-3.1-8B-Instruct backend, EPIC outperforms NV-Embed-v2 by 18.79%p on average across the four datasets. We attribute

these gains to (i) removing preference-irrelevant noise via semantic-based coarse filtering, (ii) preserving and amplifying preference-relevant content via preference-aligned fine verification, and (iii) reusing the same preference signal with preference-guided query steering at retrieval time.

### 5.3. Cost Analysis

Figure 3 highlights the efficiency of EPIC along three axes. First, EPIC achieves a substantially smaller on-disk memory usage (Figure 3a) by retaining only preference-relevant items and indexing compact instructions rather than raw text or large auxiliary structures. Second, EPIC maintains low end-to-end retrieval latency (Figure 3b): although query steering adds a small constant overhead, retrieval remains a single FAISS kNN search over a much smaller index. This yields consistently lower latency than preference-conditioned RAG baselines (e.g., Pref-QR and PBR), which must invoke an LLM to rewrite or expand the query before retrieval. Third, EPIC reduces indexing latency (Figure 3c) compared to LLM-augmented indexing methods and even query-expansion baselines: coarse-grained filtering sharply limits the number of items passed to LM processing, enabling efficient construction without sacrificing preference-following accuracy. Measurement details are provided in Appendix B.5.

### 5.4. On-Device Experiment

**Setup.** To assess practical deployability under realistic on-device constraints, we evaluate EPIC across three edge and consumer-level platforms: a Jetson Orin Nano, a MacBook Pro M4, and a Galaxy Z Flip 6. We use these plat-

forms to measure component-wise retrieval and indexing latency. We then use the Jetson Orin Nano for the streaming robustness study on PrefWiki, where the agent continuously observes new items over time and incrementally updates its on-device memory while the active preference profile stochastically changes to model preference drift (see Figure 5 in Appendix B.6). This streaming setup serves as a dynamic evaluation protocol for on-device preference memory, testing whether EPIC can maintain preference-aligned retrieval and response grounding as both incoming data and active preferences evolve over time, without rebuilding the index from scratch.

**Runtime overhead.** We analyze the runtime overhead of EPIC to assess whether it remains practically deployable under on-device resource constraints. Table 3 reports a component-wise breakdown of retrieval latency per query and indexing latency per incoming item across the three evaluated platforms. Across these devices, EPIC achieves low end-to-end retrieval latency, ranging from **5.21 ms** on the Galaxy Z Flip 6 to **29.35 ms** on the Jetson Orin Nano. Despite incorporating *preference-guided query steering* at retrieval time, the steering overhead remains marginal across devices (ranging from **0.03** to **0.55 ms**), and FAISS retrieval also contributes only a small fraction of the total retrieval cost. During indexing, EPIC first applies fast *semantic-based coarse filtering* and then performs *preference-aligned fine verification* only for a small retained subset. As a result, fine verification invokes the LLM only 0.22 times per item on average, while total indexing latency remains within **51.42** to **249.76 ms** per incoming item across devices. Detailed device specifications and execution settings are provided in Appendix B.6. Overall, these results indicate that EPIC can be maintained online with manageable latency across both edge and consumer-level on-device platforms.

**Robustness under preference drift.** User preferences are not static: they may drift over time or change with context (e.g., budget, health, location), and a practical on-device memory should adapt without rebuilding from scratch. EPIC supports such updates by adding or removing preference embeddings and refining retrieval accordingly. In our streaming evaluation on the Jetson Orin Nano, where items accumulate while the active preference profile stochastically drifts, EPIC maintains higher preference-following accuracy while keeping memory nearly constant (Figure 4). In contrast, Contriever exhibits steadily growing memory usage as the stream progresses.

## 5.5. Ablation Study

We perform an ablation study on three key components of EPIC. Table 4 reports preference-following accuracy and

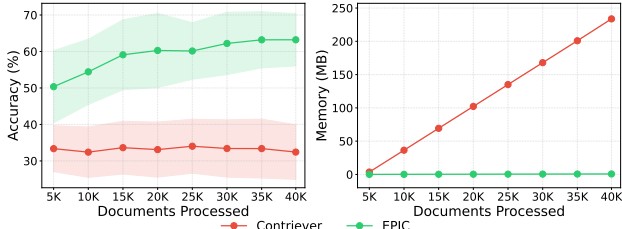

*Figure 4.* **On-device streaming data setup with random preference drift.** On Jetson Orin Nano using PrefWiki, EPIC maintains higher preference-following accuracy while keeping memory nearly constant, compared to the lightweight Contriever.

index-memory usage across PrefWiki, PrefRQ, PrefELI5, and PrefEval as modules are incrementally added. Applying semantic-based coarse filtering (**C**) alone yields the primary memory savings, reducing on-disk footprint by 3.95× to 77.68× across benchmarks by discarding preference-irrelevant items early via embedding-level matching. However, C alone does not reliably improve accuracy, suggesting that similarity-based filtering can retain partially relevant noise or miss preference-critical content. Adding preference-aligned fine verification (**C+F**) consistently improves accuracy across all benchmarks (e.g., +13.22 to +33.69%p) while further compressing memory by an additional 3.57× to 40.57× relative to C alone. This indicates that instruction-centric memory construction both strengthens preference alignment and replaces bulky raw items with compact, preference-aware representations. Finally, integrating preference-guided query steering (**C+F+S**) improves accuracy on all datasets (+0.78 to +4.03%p) without increasing memory usage, showing that steering better exploits the refined preference signals already encoded in instruction embeddings at retrieval time. Overall, coarse filtering delivers the dominant memory reduction, fine-grained instruction generation is crucial for preference alignment, and query steering provides a consistent accuracy boost at no additional storage cost.

**Applicability to structured RAG pipelines.** Beyond our default FAISS-based memory, we test whether EPIC components extend to structured RAG pipelines beyond flat vector retrieval. We integrate EPIC into HippoRAG 2 on PrefWiki with three incremental variants: **F**, **C+F**, and **C+F+S**. The results follow the main ablation trend. **F** improves preference relevance by filtering semantically plausible but weakly preference-aligned memory entries, yielding a +10.57%p accuracy gain over HippoRAG 2. Adding **C** provides the main efficiency gain by reducing the content passed to graph construction and verification, reducing memory by 6.08× and indexing latency by 8.31×. Finally, **S** further improves accuracy by +6.49%p without increasing storage. These results suggest that EPIC can serve as a plug-and-play preference-aligned memory construction layer for structured RAG pipelines. Detailed settings and results are provided in Appendix C.3.

*Table 4.* **Incremental ablation of EPIC components.** **C**: semantic-based coarse filtering (Sec. 3.1); **F**: preference-aligned fine verification (Sec. 3.2); **S**: preference-guided query steering (Sec. 3.3). Accuracy (%) and memory usage (MB) are reported on all four benchmarks using Llama-3.1-8B-Instruct.

| | | | PrefWiki | | PrefRQ | | PrefELI5 | | PrefEval | |
|---|---|---|---|---|---|---|---|---|---|---|
| **C** | **F** | **S** | Accuracy (%) | Memory Usage (MB) | Accuracy (%) | Memory Usage (MB) | Accuracy (%) | Memory Usage (MB) | Accuracy (%) | Memory Usage (MB) |
| ✗ | ✗ | ✗ | 40.88 | 142.16 | 68.11 | 142.16 | 69.45 | 133.54 | 27.89 | 59.31 |
| ✓ | ✗ | ✗ | 37.26 | 1.83 | 69.00 | 3.28 | 69.18 | 25.93 | 27.89 | 15.01 |
| ✓ | ✓ | ✗ | 52.53 | **0.27** | 82.22 | **0.92** | 85.48 | **4.50** | 61.58 | **0.37** |
| ✓ | ✓ | ✓ | **54.07** | **0.27** | **83.00** | **0.92** | **87.95** | **4.50** | **65.61** | **0.37** |

## 5.6. Threshold Sensitivity

The coarse filtering threshold $\tau$ controls the trade-off between preference-aligned retention and memory efficiency. Appendix C.4 reports sensitivity results on all four datasets using Llama-3.1-8B-Instruct. Overall, smaller thresholds retain more candidate items and can improve coverage, but they substantially increase indexing latency and memory usage. In contrast, overly large thresholds aggressively prune the memory and can reduce cost, but they may hurt accuracy when useful evidence is only weakly matched to the preference. This behavior highlights a coverage trade-off for weakly preference-matched or out-of-distribution queries, where strict filtering may discard tail evidence that is still useful for factual grounding. We therefore use $\tau = 0.3$ as an operating point for resource-constrained deployment, balancing accuracy, latency, and memory rather than optimizing a single metric. In practical systems, low query-preference similarity can trigger a broader retrieval fallback or use a less aggressively filtered memory tier.

## 6. Discussion

**Memory management.** Selecting *what* to store is only part of the on-device challenge; long-term deployment also requires policies for *how* to manage memory as it grows. While EPIC substantially mitigates growth by retaining only preference-relevant items and indexing compact instructions, accumulation is inevitable over extended use. A promising direction is to leverage the explicit preference-item associations in our memory representation to support principled retention and eviction. In addition, since the Decision Module (DM) outputs `Keep`/`Discard` decisions, the log-probabilities of these tokens are cached as a lightweight confidence score. This score can guide memory management by prioritizing high-confidence entries for long-term retention and scheduling low-confidence entries for re-verification, compression, or eviction under tight budgets. Incorporating such management mechanisms would further improve practicality, enabling sustained preference alignment and resource efficiency under tight memory budgets.

**Scope and complementarity.** EPIC addresses the *what-to-store* problem for on-device personalized RAG: it selects preference-relevant memory before indexing under tight memory and latency budgets. This scope is orthogonal to high-compression retrieval methods such as BPR (Yamada et al., 2021) and RaBitQ (Gao & Long, 2024), which compress vector representations after the candidate memory contents have already been retained; in contrast, EPIC reduces memory at the content level by deciding which items should be stored in the first place. This distinction suggests that index compression can be combined with EPIC in future systems. EPIC is also distinct from memory-side personalization methods, which typically assume that user memories, profiles, or interaction histories are already available and focus on organizing or retrieving them for personalized generation. By contrast, EPIC targets the preceding memory construction stage from raw candidate data under strict on-device constraints. Appendix C.2 and Appendix C.1 provide comparisons with high-compression retrieval and memory-side baselines, respectively.

## 7. Conclusion

We introduced EPIC, a framework for constructing *compact, preference-aligned on-device memory* from raw data under privacy and resource constraints. By filtering and storing only preference-relevant items and aligning retrieval with user intent, EPIC consistently improves preference-following accuracy while drastically reducing memory footprint and maintaining low retrieval latency across four benchmarks. Our on-device evaluation across three edge and consumer-level platforms shows that EPIC remains lightweight under realistic device constraints, while the Jetson streaming study demonstrates that it can maintain preference-aligned retrieval under streaming data and preference drift. These results suggest that preference-aligned memory construction is a promising direction for bringing personalized RAG from server-side settings to resource-constrained personal devices. Overall, EPIC provides a foundation for privacy-preserving on-device personal AI agents that must deliver high-fidelity personalization under tight device budgets.

## Acknowledgements

This work was supported by Institute of Information & Communications Technology Planning & Evaluation (IITP) grant funded by the Korea government (MSIT) (RS-2025-25442824, AI Star Fellowship Program (Ulsan National Institute of Science and Technology)); Institute of Information & Communications Technology Planning & Evaluation (IITP) grant funded by the Korea government (MSIT) (No. RS-2020-II201336, Artificial Intelligence Graduate School Program (UNIST)); and Institute of Information & Communications Technology Planning & Evaluation (IITP) grant funded by the Korea government (MSIT) (No. RS-2026-25527532, Hyper-scale Industrial AI Research Support (R&D) Program, Development of an industry-specified on-device AI technology). This work was also supported by the National Research Foundation of Korea (NRF) grant funded by the Korea government (MSIT) (RS-2025-00553241). In addition, this research was supported by the "Advanced GPU Utilization Support Program" funded by the Government of the Republic of Korea (Ministry of Science and ICT).

We also thank the UAI Lab members, especially Yeji, Ahin, and Wooyoung, for their valuable feedback and support in improving the overall quality of this paper.

## Impact Statement

This paper presents work whose goal is to advance the field of Machine Learning. There are many potential societal consequences of our work, none of which we feel must be specifically highlighted here.

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

# A. Detailed Related work

## A.1. Parametric Knowledge-Based Personalization

Mainstream alignment via instruction tuning and reinforcement learning from human feedback (RLHF) optimizes for aggregate human preferences (Ouyang et al., 2022), which can dilute or even conflict with the idiosyncratic needs of specific users. Early personalization approaches adapt model parameters through full fine-tuning or alignment with user-specific data, but these approaches are computationally and operationally expensive at scale (e.g., separate fine-tuned checkpoints per user). Parameter-efficient fine-tuning (PEFT) mitigates these costs by updating small subsets or low-rank projections while freezing most weights. Adapters (Houlsby et al., 2019) insert lightweight modules between transformer layers; LoRA (Hu et al., 2022) injects trainable low-rank matrices and reports up to 10,000× fewer trainable parameters and 3× lower memory versus full fine-tuning on GPT-3-class models; prefix-tuning (Li & Liang, 2021) and prompt-tuning (Lester et al., 2021) learn continuous "soft prompts" that steer frozen backbones. While effective, these techniques still require per-user artifacts (raising storage, routing, and lifecycle overhead), can suffer from forgetting or entanglement under continual updates, and may pose privacy concerns when centralizing user-specific gradients or weights. Recent work explores explicitly personalized PEFT and privacy-aware variants. One PEFT Per User (OPPU) (Tan et al., 2024) attaches user-specific PEFT modules that can be plugged into a shared base model and combined with non-parametric user profiles, improving personalization while preserving model ownership and reducing central data exposure. In contrast, our work pursues non-parametric on-device personalization by constructing compact instruction-indexed memory, avoiding per-user weight updates while enabling continual updates from streaming logs.

## A.2. Memory-Side Personalization

Memory-side personalization methods construct retrieval memory from user-specific artifacts, such as user-authored documents, profiles, editable memories, or curated interaction histories. These methods are closely related to EPIC in that they modify the memory side of RAG rather than only rewriting the query. However, most of them assume that useful personal artifacts or memories are already available and focus on organizing or retrieving from such memories.

In contrast, EPIC addresses an earlier stage of the pipeline: deciding what to retain from raw, heterogeneous, and continuously growing on-device data under strict memory constraints. Thus, memory-side personalization methods are complementary to EPIC rather than direct replacements. When applicable, EPIC can serve as a preference-aligned memory construction layer before such memory-side retrieval methods are applied. We provide additional baseline discussion and analysis in Appendix C.1.

## A.3. Vector Quantization and Index Compression

Vector quantization and product quantization reduce the storage cost of dense vector indexes by compressing embedding vectors after documents have already been selected and encoded. This line of work is highly relevant to on-device RAG because it improves the storage efficiency of retrieval indexes.

However, vector quantization addresses a different stage from EPIC. Quantization optimizes how to store vectors, whereas EPIC optimizes what to store in the first place. Therefore, quantization can reduce the size of a dense index, but it does not remove preference-irrelevant documents, raw text, or auxiliary retrieval structures. In principle, EPIC and vector quantization are orthogonal and can be combined. We provide quantitative comparisons with vector-quantization baselines in Appendix C.2.

## A.4. User Preference Dataset

User Preference Datasets collect examples in which a user's tastes, constraints, or style (e.g., likes/dislikes, tone, format, accessibility needs) are stated or implied, and models are evaluated on whether they respect those preferences in their responses. Early work centered on persona-conditioned dialogue (e.g., profile sentences guiding open-domain conversation). More recent researches include longer-context interactions where a model must infer, remember, and apply preferences over multiple turns. While valuable for personalized generation, most such datasets were not designed to directly test retrieval over an external corpus. Personalized RAG must retrieve documents that satisfy both the information need query and the user's preferences, then ground the answer on those documents. Existing user-preference datasets rarely support

rigorous evaluation of this retrieval objective for several reasons:

1. Either the user preference or the question is missing, so the retrieval target cannot be precisely defined.

2. Questions rarely induce preference conflicts, making violations unlikely and the retrieval task non-discriminative.

3. No gold labels tying (preference, question) pairs to documents that both answer the query and satisfy preferences.

In light of these limitations of existing datasets, this study makes extensive use of the PrefEval benchmark (Zhao et al., 2025).

### A.5. PrefEval Benchmark

The Explicit Preference subset of PrefEval dataset (Zhao et al., 2025) focuses on preferences the user states unambiguously (e.g., "I avoid electric vehicles," "I prefer spicy food"). Instances typically pair:

1. a preference statement (clear like/dislike or constraint), and

2. a query that can easily elicit a default answer which would violate that preference unless the model takes it into account (e.g., recommending the best compact cars for city driving, where the most top options are electric vehicles),

3. optionally, a short explanation/rationale highlighting why the query is risky with respect to the preference.

This subset deliberately booby-traps the obvious answer: the quickest generic response is often preference-inconsistent. Strong performance therefore requires the model to (1) recognize the explicit constraint, (2) prioritize it alongside topical relevance, and (3) surface alternatives that respect the constraint.

The four error types are: (1) Preference-Unaware Violation: The LLM provides generic recommendations that contradict the user's stated preference due to unawareness of user preference. (2) Preference Hallucination Violation: The response fabricates or misattributes preferences, diverging from the user's true preference and violates the true preference. (3) Inconsistent Violation: The response acknowledges the correct preference but generates contradicting response. (4) Unhelpful Response: The response lacks relevant recommendations or fails to address the query due to poor recall of the user's preference. To validate our LLM-based evaluation method, we manually checked 200 randomly sampled evaluations, with an observed 5% error rate. This demonstrates strong agreement between human judgment and LLM-based assessments with Claude 3 Sonnet.

Importantly, a notable aspect of PrefEval is its evaluation methodology, which leverages an LLM-based judge to categorize errors in preference following. Instead of relying solely on BLEU/ROUGE or costly human ratings, PrefEval uses an LLM to automatically check each generated response against the user's stated preference. The evaluation defines four possible error types (failure modes), aggregated from binary criteria:

1. Preference-Unaware Violation: The LLM provides generic recommendations that contradict the user's stated preference due to unawareness of user preference.

2. Preference Hallucination Violation: The response fabricates or misattributes preferences, diverging from the user's true preference and violates the true preference.

3. Inconsistent Violation: The response acknowledges the correct preference but generates contradicting response.

4. Unhelpful Response: The response lacks relevant recommendations or fails to address the query due to poor recall of the user's preference.

# B. Experimental Details

## B.1. Corpus of Preference Benchmarks

This section describes the retrieval corpora used for indexing and retrieval. Since each source corpus has a different structure, we sample at the article, document, or chunk level depending on the benchmark, and then use the resulting chunked retrieval corpus consistently across all methods. Dataset construction details, including preference-question generation and persona construction, are provided separately in Appendix D.

**Wikipedia corpus**    PrefWiki and PrefRQ rely on the English Wikipedia as the underlying retrieval corpus. We use the official dump released on 2025-04-04 (`enwiki/latest`).[1] Following Chen et al. (2017), we process the dump using the `WikiExtractor` script[2], which removes MediaWiki markup and retains only plain text. Each processed article is stored in JSON format with two fields: `title` and `text`. The extracted snapshot contains 6,945,964 documents totaling 17.74 GB of plaintext. For index construction in our experiments, we uniformly sample 10,000 articles and segment them into retrieval units ("chunks"), resulting in 39920 chunks. This chunked corpus is used consistently across PrefWiki, and PrefRQ evaluations.

**ELI5 corpus**    From the full collection of supporting documents provided by the ELI5 dataset, which comprises 16,453,150 documents (approximately 185 GB), we randomly sample 2,000 documents. These are preprocessed and segmented into retrieval units ("chunks") following the same procedure applied to the Wikipedia corpus. This yields a total of 37,786 chunks, which is comparable in scale to the chunked Wikipedia corpus. The resulting chunked corpus is employed exclusively for the PrefELI5 evaluation.

**LMSYS-Chat corpus**    From the LMSYS-Chat-1M dataset (`lmsys/lmsys-chat-1m`), we construct a conversation corpus by grouping examples by `conversation_id` and retaining the snapshot with the largest `turn` value for each conversation. We serialize each conversation into plain text using the format "`role:  content`" and segment it into overlapping retrieval units with a sliding-window chunking strategy (max 8 turns and ∼180 words per chunk, with 2-turn overlap). From the resulting chunk pool, we uniformly sample 10,000 chunks to form the retrieval corpus used in PrefEval.

## B.2. Chunking Strategy

In retrieval-augmented generation, source texts must be divided into smaller segments to enable precise retrieval. Without chunking, retrieval systems risk pulling in overly broad or irrelevant sections, thereby diminishing contextual alignment and response quality. In our experiments, we implemented a fixed-size chunking strategy with semantic safeguards. Source documents were first segmented into chunks of approximately 100 words. To preserve coherence, when a single sentence exceeded the 100-word threshold, we retained the entire sentence within the chunk rather than dividing it. This approach ensures that semantic integrity and contextual continuity are not compromised by arbitrary truncation. By structuring the data in this way, each chunk becomes a self-contained unit of meaning, allowing the retrieval system to assess its relevance independently. Importantly, this strategy was not only applied to our proposed method, EPIC, but also consistently enforced across all baseline models in our experiments. By adopting an identical preprocessing pipeline for every system under evaluation, we ensured that performance comparisons reflect genuine methodological differences rather than artifacts of data segmentation.

## B.3. Models, Retrieval, and Inference

**Server-side inference and evaluation.**    For large-scale benchmark comparisons and offline analyses, we evaluate three instruction-tuned LLMs of different scales: Llama-3.1-8B-Instruct, gpt-oss-20b, and Qwen3-4B-Instruct-2507. All server-side LLM inference is conducted using the `vLLM` engine (Kwon et al., 2023), which provides optimized memory management and high-throughput serving. For LLM-based evaluation, we use LLaMA-3.3-70B-Instruct following the PrefEval evaluation protocol. To ensure reproducibility, we fix the random seed to `0`, set the generation temperature to `0.0`, and use `float32` computation unless otherwise specified.

---

[1] https://dumps.wikimedia.org/enwiki/latest/
[2] https://github.com/attardi/wikiextractor

**Retrieval and indexing setup.** Unless otherwise specified by a baseline, we encode chunks, preferences, and queries using Contriever embeddings with dimension 768. All dense embeddings are $\ell_2$-normalized, so inner-product search is equivalent to cosine similarity. Vector indexing and nearest-neighbor search are implemented with FAISS (Johnson et al., 2019), using the `IndexFlatIP` backend for exact inner-product kNN search. We use the same retrieval depth $k = 5$ across all methods unless a baseline requires its own retrieval configuration.

**Device-side inference.** The on-device streaming and latency experiments are conducted separately from the server-side `vLLM` setup. These experiments run locally on each target device using device-specific inference backends and quantized on-device execution. Accordingly, their latency, memory footprint, and streaming-update measurements are not obtained from the server-side environment. Detailed device-specific execution settings are provided in Appendix B.6.

## B.4. Algorithmic Implementation of EPIC

Algorithm 1 summarizes the implementation of EPIC. The procedure consists of three stages corresponding to Semantic-Based Coarse Filtering (Sec. 3.1), Preference-Aligned Fine Verification (Sec. 3.2), and Preference-Guided Query Steering (Sec. 3.3).

---

**Algorithm 1** EPIC memory construction and preference-guided retrieval.

---

**Input:** Candidate chunks $\mathcal{D}$, user preferences $\mathcal{P}$, encoder $\mathrm{E}(\cdot)$, threshold $\tau$, top-$K$ value $K$, query $q$
**Output:** Preference-aligned memory $\mathcal{M}$ and retrieved context $\mathcal{C}_q$

1: **Stage 1: Semantic-Based Coarse Filtering** *Sec. 3.1*
2: Compute preference embeddings $\{\mathrm{E}(p) \mid p \in \mathcal{P}\}$
3: Initialize coarse candidate set $\mathcal{D}_{\mathrm{coarse}} \leftarrow \emptyset$
4: **for** each chunk $x \in \mathcal{D}$ **do**
5:     Compute chunk embedding $\mathrm{E}(x)$
6:     $\mathcal{P}_{\mathrm{rel}}(x) \leftarrow \{p \in \mathcal{P} \mid \mathrm{Sim}(\mathrm{E}(x), \mathrm{E}(p)) \geq \tau\}$
7:     **if** $\mathcal{P}_{\mathrm{rel}}(x) \neq \emptyset$ **then**
8:         Add $x$ to $\mathcal{D}_{\mathrm{coarse}}$
9:     **end if**
10: **end for**
11: **Stage 2: Preference-Aligned Fine Verification** *Sec. 3.2*
12: Initialize preference-aligned memory $\mathcal{M} \leftarrow \emptyset$
13: **for** each chunk $x \in \mathcal{D}_{\mathrm{coarse}}$ **do**
14:     $(d, r, \mathcal{P}'_{\mathrm{rel}}(x)) \leftarrow \mathrm{DM}(x, \mathcal{P}_{\mathrm{rel}}(x))$
15:     **if** $d = \langle \mathrm{Discard} \rangle$ **then**
16:         **continue**
17:     **end if**
18:     **for** each preference $p \in \mathcal{P}'_{\mathrm{rel}}(x)$ **do**
19:         $i_x(p) \leftarrow \mathrm{IG}(x, p, r)$
20:         Add $(x, i_x(p), p, \mathrm{E}(i_x(p)))$ to $\mathcal{M}$
21:     **end for**
22: **end for**
23: Build a FAISS index $\mathcal{F}$ over $\{\mathrm{E}(i) \mid (x, i, p, \mathrm{E}(i)) \in \mathcal{M}\}$
24: **Stage 3: Preference-Guided Query Steering** *Sec. 3.3*
25: Compute query embedding $\mathrm{E}(q)$
26: $p^* \leftarrow \arg\max_{p \in \mathcal{P}} \mathrm{Sim}(\mathrm{E}(q), \mathrm{E}(p))$
27: $\tilde{q} \leftarrow \frac{\mathrm{E}(q) + \mathrm{E}(p^*)}{\| \mathrm{E}(q) + \mathrm{E}(p^*) \|_2}$
28: Retrieve top-$K$ entries $\mathcal{R}_q$ from $\mathcal{F}$ using $\tilde{q}$
29: $\mathcal{C}_q \leftarrow \{x \mid (x, i, p, \mathrm{E}(i)) \in \mathcal{R}_q\}$
30: **return** $\mathcal{M}, \mathcal{C}_q$

---

## B.5. Server-Side Evaluation

**Environment.** Unless otherwise stated, the large-scale benchmark comparisons and offline analyses are run on a single server with $2 \times$ AMD EPYC 9354 CPUs (32 cores / 64 threads each; 64 cores / 128 threads total) and $8 \times$ NVIDIA RTX 6000 Ada GPUs (48 GB each). This server environment is used for fair and reproducible comparisons across baselines and EPIC under the same hardware and software configuration.

**Memory usage measurement.** Table 5 presents the memory usage requirements (in MB) for various Retrieval-Augmented Generation (RAG) methods across different indexing strategies. The breakdown highlights the memory consumed by raw document storage, FAISS index, RAPTOR tree, HippoRAG graph structure, embeddings, and additional miscellaneous components. For EPIC, the miscellaneous category includes the stored preference-aligned instruction (0.04 MB) and the preference embedding (0.03 MB).

**Retrieval latency measurement.** We measure retrieval latency as the end-to-end wall-clock time per query required to obtain the final retrieved context (excluding downstream answer generation). For sparse retrieval (BM25), latency corresponds to the time to score and retrieve top-$k$ passages from the raw text corpus using a BM25 index. For standard dense RAG baselines (e.g., Contriever and NV-Embed-v2), we measure FAISS top-$k$ search time over chunk embeddings.

For indexing-enhanced methods, we measure the full method-specific retrieval procedure, including structured traversal and any additional scoring stages (e.g., RAPTOR tree traversal/selection and HippoRAG/HippoRAG 2 graph-based retrieval). For preference-conditioned methods, we include query-time personalization overheads prior to retrieval. Specifically, Pref-QR includes the query rewriting time. For PBR, following the original pipeline, we include the time for generating *Pseudo Utterance* and *Pseudo Reasoning* used to condition retrieval, in addition to the subsequent retrieval time. All results are averaged over evaluation queries under identical hardware and batching settings.

**Indexing latency measurement.** Table 7 reports the end-to-end indexing latency (in seconds) on PrefWiki using Llama-3.1-8B-Instruct under different RAG indexing strategies. We decompose the latency into (i) embedding computation for indexable units, (ii) FAISS index construction, (iii) method-specific structure construction (e.g., trees or graphs), and (iv) additional processing such as information extraction or LLM-based filtering, when applicable.

Standard dense RAG baselines primarily incur embedding time, while BM25 requires no embedding-based indexing. Indexing-enhanced approaches introduce substantial overhead for building structured memories (e.g., RAPTOR's hierarchical tree construction and HippoRAG-style graph construction), which can dominate total latency. Preference-conditioned methods add extra structure-building steps (e.g., memory graph building in PBR). In contrast, EPIC maintains low indexing overhead by applying lightweight cosine-based filtering and a small amount of LLM-based refinement and instruction generation, followed by a minimal FAISS build step.

*Table 5.* Analysis on Memory Usage. PrefWiki, Llama-3.1-8B-Instruct

| | Method | Raw Documents | FAISS Index | RAPTOR Tree | Graph Memory | Embeddings | Misc. | Total |
|---|---|---|---|---|---|---|---|---|
| | *Standard RAG* | | | | | | | |
| | BM25 (Robertson et al., 1995) | 25.21 | 0 | 0 | 0 | 0 | 0 | 25.21 |
| | Contriever (Izacard et al., 2022) | 25.21 | 116.95 | 0 | 0 | 0 | 0 | 142.16 |
| | NV-Embed-v2 (Lee et al., 2025) | 25.21 | 623.75 | 0 | 0 | 0 | 0 | 648.96 |
| | *Indexing-enhanced RAG* | | | | | | | |
| Memory Usage (MB) | RAPTOR (Sarthi et al., 2024) | 0 | 0 | 297.05 | 0 | 0 | 0 | 297.05 |
| | HippoRAG (Gutiérrez et al., 2024) | 26.91 | 0 | 0 | 1.54 | 2282.44 | 36.57 | 2347.46 |
| | HippoRAG 2 (Gutiérrez et al., 2025) | 0 | 0 | 0 | 130.74 | 2765.76 | 0 | 2896.50 |
| | *Preference-conditioned RAG* | | | | | | | |
| | Pref-QR (Zhou et al., 2024) | 25.21 | 116.95 | 0 | 0 | 0 | 0 | 142.16 |
| | PBR (Zhang et al., 2026) | 25.21 | 116.95 | 0 | 144.32 | 0 | 0 | 286.48 |
| | **EPIC** | 0.05 | 0.15 | 0 | 0 | 0 | 0.07 | 0.27 |

*Table 6.* Analysis on Retrieval Latency. four benchmarks, Llama-3.1-8B-Instruct

|  | Method | PrefWiki | PrefRQ | PrefELI5 | PrefEval | **Average** |
|---|---|---|---|---|---|---|
|  | *Standard RAG* | | | | | |
|  | BM25 (Robertson et al., 1995) | 99 | 62 | 355 | 43 | 139.75 |
|  | Contriever (Izacard et al., 2022) | 6 | 6 | 6 | 6 | 6 |
|  | NV-Embed-v2 (Lee et al., 2025) | 100 | 100 | 102 | 84 | 96.5 |
|  | *Indexing-enhanced RAG* | | | | | |
|  | RAPTOR (Sarthi et al., 2024) | 360 | 363 | 369 | 127 | 304.75 |
|  | HippoRAG (Gutiérrez et al., 2024) | 343 | 478 | 398 | 580 | 449.75 |
|  | HippoRAG 2 (Gutiérrez et al., 2025) | 812 | 364 | 388 | 896 | 615.00 |
|  | *Preference-conditioned RAG* | | | | | |
|  | Pref-QR (Zhou et al., 2024) | 513 | 444 | 798 | 592 | 586.75 |
|  | PBR (Zhang et al., 2026) | 10,073 | 10,219 | 15,251 | 17,765 | 13,327 |
|  | **EPIC** | 3 | 3 | 3 | 3 | 3 |

*Retrieval Latency (ms)*

*Table 7.* Analysis on Indexing Latency (seconds). PrefWiki, Llama-3.1-8B-Instruct. Indexing Latency is measured as wall-clock time to construct the retrieval index and any auxiliary structures from the full corpus.

|  | Method | Embedding | FAISS Build | Structure Build | LLM / IE | Save | Total |
|---|---|---|---|---|---|---|---|
|  | *Standard RAG* | | | | | | |
|  | BM25 (Robertson et al., 1995) | 0.00 | 0.00 | 0.00 | 0.00 | 0.00 | 0.00 |
|  | Contriever (Izacard et al., 2022) | 88.47 | 0.07 | 0.00 | 0.00 | 0.00 | **88.54** |
|  | NV-Embed-v2 (Lee et al., 2025) | 1918.69 | 0.07 | 0.00 | 0.00 | 0.00 | 1918.76 |
|  | *Indexing-enhanced RAG* | | | | | | |
|  | RAPTOR (Sarthi et al., 2024) | 0.00 | 0.00 | 15675.62 | 0.00 | 0.29 | **15676.06** |
|  | HippoRAG (Gutiérrez et al., 2024) | 0.00 | 0.00 | 45.49 | 1524.69 | 0.00 | **1568.83** |
|  | HippoRAG 2 (Gutiérrez et al., 2025) | 47.07 | 0.00 | 99.57 | 4650.89 | 0.00 | **4726.25** |
|  | *Preference-conditioned RAG* | | | | | | |
|  | Pref-QR (Zhou et al., 2024) | 88.47 | 0.07 | 0.00 | 0.00 | 0.00 | **88.54** |
|  | PBR (Zhang et al., 2026) | 123.78 | 0.07 | 529.71 | 0.00 | 0.44 | **654.00** |
|  | **EPIC** | 88.47 | 0.07 | 0.05 | 157.76 | 0.00 | **246.38** |

*Indexing Latency (s)*

## B.6. On-Device Evaluation

We evaluate EPIC entirely on-device along two complementary axes. First, the *on-device streaming evaluation* examines whether EPIC remains effective in a realistic streaming setting where new documents are continuously appended over time. We conduct two experiments under this setting on a Jetson Orin Nano rather than on the server: a base streaming evaluation with a fixed user preference profile, and an extended evaluation under *dynamic user preferences* that drift during streaming. Second, the *on-device latency evaluation* assesses the practical usability and deployability of EPIC on heterogeneous consumer hardware, for which we report a component-wise retrieval and indexing latency breakdown (Table 3) across three devices: a Jetson Orin Nano, a MacBook Pro M4, and a Galaxy Z Flip 6. All device-side measurements (retrieval latency, indexing latency, memory footprint, and streaming-update results) are obtained locally on the corresponding device with no offloading to a server. The per-device hardware specifications are summarized in Table 8.

*Table 8.* Evaluation device specifications.

|  | **Jetson Orin Nano** | **MacBook Pro M4** | **Galaxy Z Flip 6** |
|---|---|---|---|
| Type | Edge AI module | Laptop | Smartphone |
| Release | Jan 2023 | Oct 2024 | Jul 2024 |
| CPU | 6-core Cortex-A78AE | 10-core Apple M4 | 8-core Snapdragon 8 Gen 3 |
| GPU | Ampere GPU | 10-core M4 GPU | Adreno 750 |
| NPU | – | 16-core Apple Neural Engine | Hexagon NPU V75 |
| Memory | 8GB LPDDR5 | 16GB unified memory | 12GB LPDDR5X |

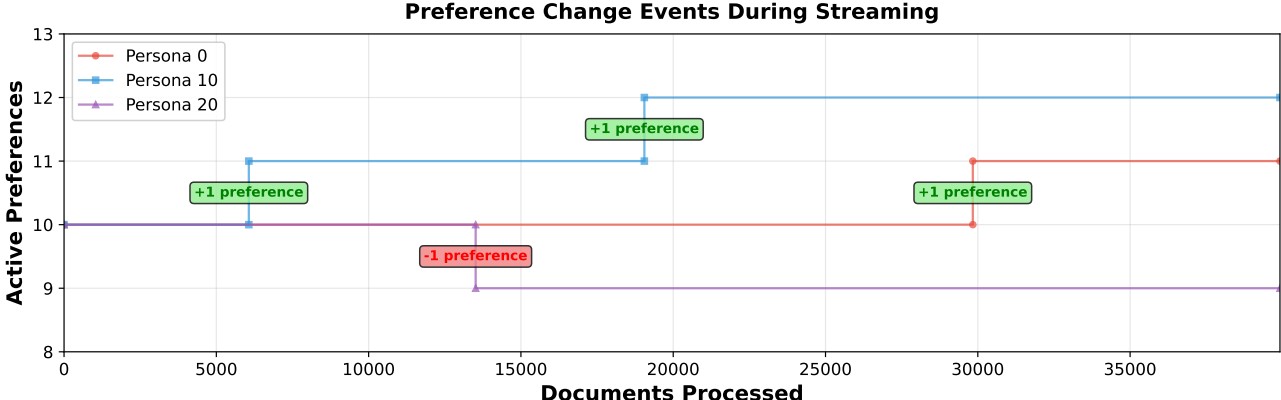

*Figure 5.* **Preference change events during streaming (examples).** To complement Figure 4, we visualize stochastic preference drift by plotting the number of active preferences over time for three representative personas (0, 10, and 20). Markers indicate randomized preference update events (e.g., +1/-1 preference) applied online while the document stream continues.

**On-device streaming evaluation with dynamic preferences.** We evaluate practical on-device feasibility on NVIDIA Jetson Orin Nano using a 4-bit Qwen3-4B model to reflect realistic deployment constraints. We model a streaming device setting where the agent continuously encounters and appends new documents over time. We partition the stream into fixed-size batches and perform evaluation every 5,000 newly observed documents. At each *streaming step* (i.e., every 5,000 documents), the system incrementally updates its on-device memory from the newly appended batch and then answers a fixed set of evaluation queries.

**Dynamic preference evaluation.** Under the same on-device streaming setting, we model preference drift by introducing stochastic preference changes during streaming. Concretely, with a fixed probability per step, the active user preference profile is randomly updated (e.g., add/remove of preferences) while the incoming document stream continues. This setting tests robustness to non-stationary user behavior. Figure 5 visualizes representative preference-change traces for three personas (0, 10, and 20), while Figure 4 aggregates performance under such stochastic drift across all personas.

**On-device latency evaluation.** To assess practical deployability across heterogeneous on-device platforms, we measure EPIC on three devices: a Jetson Orin Nano, a MacBook Pro M4, and a Galaxy Z Flip 6. The component-wise latency breakdown for these devices is reported in Table 3; here we detail the corresponding execution settings. Across all three platforms, we use the same models: the LLM is Qwen3-4B-Instruct-2507 quantized to 4-bit, and the embedding model is `facebook/contriever` executed in FP16 precision. The execution backend for each component, however, is tailored to the accelerators available on each device:

- **Jetson Orin Nano:** As the device has no dedicated NPU, both LLM inference and Contriever embeddings are GPU-accelerated through CUDA.

- **MacBook Pro M4:** LLM inference is GPU-accelerated through `mlx_lm` on Apple Silicon, while Contriever embeddings run on the Apple Neural Engine via a CoreML-converted model.

- **Galaxy Z Flip 6:** LLM inference is served through a local `llama.cpp` API with OpenCL-based GPU acceleration, while Contriever embeddings run on the Qualcomm Hexagon NPU through an ExecuTorch/QNN backend.

# C. Additional Baselines and Analyses

## C.1. Memory-Side Personalization Baselines

Building on the discussion in Appendix A.2, we additionally compare EPIC with a memory-side personalization baseline. Specifically, we implement MemoryBank (Zhong et al., 2024), which maintains long-term user memory and retrieves relevant memories for personalized generation. Since MemoryBank is designed for conversation-based memory, we conduct this comparison on PrefEval, which is the conversation-history benchmark in our evaluation suite. Results are reported using Llama-3.1-8B-Instruct and averaged over 57 profiles (persona IDs 0-56).

*Table 9.* Comparison with a memory-side personalization baseline on PrefEval. MemoryBank is evaluated on PrefEval because it is designed for conversation-based long-term memory.

| Method | Accuracy (%) | Indexing Latency (s) | Retrieval Latency (ms) | Memory (MB) |
|---|---|---|---|---|
| MemoryBank (Zhong et al., 2024) | 25.61 | 4338.76 | 9.0 | 36.38 |
| **EPIC (ours)** | **64.0** | **162.16** | **3.0** | **0.41** |

The results show that EPIC substantially outperforms MemoryBank in both preference-following accuracy and efficiency. While MemoryBank stores and retrieves long-term conversational memories, it does not explicitly optimize what should be retained under strict on-device memory constraints. In contrast, EPIC constructs compact preference-aligned memory by filtering and verifying preference-relevant content before indexing.

## C.2. High-Compression Retrieval Baselines

Building on Appendix A.3, we compare EPIC with high-compression retrieval baselines to distinguish preference-aligned memory construction from post-hoc index compression. We implement BPR (Yamada et al., 2021) for hashing-based binary-code retrieval and RaBitQ (Gao & Long, 2024) for vector quantization. These methods compress the representation of the dense index after chunks have already been retained, whereas EPIC reduces memory by deciding what to store before index construction. We evaluate all methods on PrefWiki using Llama-3.1-8B-Instruct and average results over 57 profiles (persona IDs 0-56).

*Table 10.* Comparison with high-compression retrieval baselines on PrefWiki. Compression ratios are computed relative to the Contriever baseline memory footprint of 142.16 MB.

| Method | Accuracy (%) | Indexing Latency (s) | Retrieval Latency (ms) | Memory (MB) |
|---|---|---|---|---|
| BPR (Yamada et al., 2021) | 41.33 | **88.85** | 24.10 | 28.86 (4.93×) |
| RaBitQ (Gao & Long, 2024) | 40.77 | 91.34 | **0.95** | 30.06 (4.73×) |
| **EPIC (ours)** | **55.2** | 246.38 | 3.00 | **0.27 (526.52×)** |

BPR and RaBitQ reduce the memory footprint by compressing the dense vector index, but the retained chunk storage remains unchanged. In our setup, BPR and RaBitQ mainly reduce the vector index size while still storing the raw chunks, resulting in 28.86 MB and 30.06 MB total memory, respectively. In contrast, EPIC changes the retained memory set itself by storing only preference-relevant instruction-item pairs, reducing both the stored content and the retrieval burden. This result highlights that EPIC addresses the fundamental what-to-store problem, which is orthogonal to post-hoc index compression.

## C.3. Plug-and-Play with HippoRAG 2

To test whether the components of EPIC are tied to a flat FAISS-based retrieval pipeline, we apply them to HippoRAG 2 (Gutiérrez et al., 2025) in a plug-and-play manner. We evaluate three variants on PrefWiki using Llama-3.1-8B-Instruct: HippoRAG 2 + **F** applies preference-aligned fine verification, HippoRAG 2 + **C** + **F** applies semantic-based coarse filtering before fine verification, and HippoRAG 2 + **C** + **F** + **S** additionally applies preference-guided query steering at retrieval time. Results are averaged over 57 profiles (persona IDs 0-56).

*Table 11.* Plug-and-play results with HippoRAG 2 on PrefWiki. **C**: semantic-based coarse filtering; **F**: preference-aligned fine verification; **S**: preference-guided query steering.

| Method | Accuracy (%) | Indexing Latency (s) | Retrieval Latency (ms) | Memory (MB) |
|---|---|---|---|---|
| HippoRAG 2 (Gutiérrez et al., 2025) | 42.9 | 4726.25 | 615.00 | 2896.50 |
| HippoRAG 2 + F | 53.47 | 2215.60 | 304.32 | 22.79 |
| HippoRAG 2 + C + F | 55.26 | **266.47** | **282.99** | **3.75** |
| HippoRAG 2 + C + F + S | **61.75** | 292.42 | 306.85 | **3.75** |

These results show that the components of EPIC are complementary to graph-based RAG methods. Fine verification improves preference-following accuracy by removing semantically misleading or weakly relevant memory entries. Coarse filtering provides the main efficiency gain by reducing the number of chunks passed to graph construction and verification. Query steering further improves retrieval accuracy without increasing memory usage or indexing latency. Overall, EPIC can serve as a plug-and-play preference-aligned memory construction layer for structured RAG pipelines such as HippoRAG 2.

## C.4. Threshold Sensitivity

Table 12 reports the full threshold sensitivity results for $\tau \in \{0.0, 0.2, 0.3, 0.4\}$. All values are averaged over all personas in each dataset using Llama-3.1-8B-Instruct. We omit $\tau = 0.1$ from the table because it retained the same memory entries as $\tau = 0.0$ in our experiments, resulting in identical accuracy, latency, and memory usage.

*Table 12.* Threshold sensitivity analysis. Lat. denotes the LLM-based fine verification time (in seconds) during indexing, excluding the shared embedding cost, and Mem. denotes memory usage in MB. Bold values indicate the default operating point used in our experiments, not necessarily the best value for each metric.

| $\tau$ | PrefWiki | | | PrefRQ | | | PrefELI5 | | | PrefEval | | |
|---|---|---|---|---|---|---|---|---|---|---|---|---|
| | Acc. (%) | Lat. (s) | Mem. (MB) | Acc. (%) | Lat. (s) | Mem. (MB) | Acc. (%) | Lat. (s) | Mem. (MB) | Acc. (%) | Lat. (s) | Mem. (MB) |
| 0.0 | 60.91 | 2293.90 | 1.61 | 85.44 | 2369.23 | 6.29 | 88.49 | 2491.56 | 13.85 | 74.21 | 798.57 | 2.47 |
| 0.2 | 71.68 | 1209.05 | 1.89 | 84.56 | 1353.14 | 6.12 | 87.26 | 2195.66 | 12.27 | 77.50 | 594.41 | 1.78 |
| **0.3** | **54.07** | **157.80** | **0.27** | **83.00** | **81.90** | **0.92** | **88.00** | **605.70** | **4.50** | **65.61** | **162.20** | **0.37** |
| 0.4 | 48.20 | 1.45 | 0.01 | 73.22 | 2.66 | 0.02 | 82.33 | 14.58 | 0.24 | 42.81 | 4.70 | 0.02 |

# D. Dataset Construction Details

## D.1. Dataset-Specific Construction

**PrefWiki (Static recommendation over Wikipedia).**  PrefWiki evaluates preference-aligned retrieval over a large static knowledge corpus. We use the English Wikipedia dump released on 2025-04-04 as the underlying corpus (Wikimedia Foundation, 2025). We start from the pool of 1,000 explicit preferences provided by PrefEval (Zhao et al., 2025), and retain 583 preferences that can be reasonably mapped to Wikipedia entities or topics. Given each retained preference, we generate suggestion-style user questions using an adapted PrefEval question-generation prompt, and validate the resulting preference-question pairs with the same preference-centric assessment protocol (Appendix D.2). The validated pairs are grouped into personas as described in Appendix D.3.

**PrefRQ (Subjective debate questions with generated preferences).**  PrefRQ targets subjective, value-laden queries where preference alignment is particularly critical. We take questions from the Researchy Questions (RQ) dataset (Rosset et al., 2025) as the fixed question source, and generate an explicit preference for each question using an adapted PrefEval prompt (Zhao et al., 2025). If a generated preference-question pair is deemed invalid under our validity assessment (Appendix D.2), we regenerate the preference; after repeated failures, the question is excluded. This yields high-subjectivity questions paired with explicit preferences suitable for evaluating preference-aligned retrieval and response generation.

**PrefELI5 (Noisy web-derived digital footprints with ELI5 questions).**  PrefELI5 models noisy, long-form web footprints where raw items are verbose and weakly structured. We use questions from ELI5 (Fan et al., 2019) as the fixed question source and generate an explicit preference for each question using adapted PrefEval prompts, followed by preference-question validity assessment (Appendix D.2). Supporting documents are retrieved from a large-scale web crawl (Common Crawl, Aug. 2018), forming a noisy candidate pool that stress-tests whether EPIC can compress raw footprints into compact, preference-faithful instructions while preserving retrievability.

**PrefEval (Conversation histories).**  For conversation histories, we use PrefEval (Zhao et al., 2025), which is constructed from LMSYS-Chat-1M (Zheng et al., 2023) and includes explicit preference statements paired with user queries. This benchmark evaluates preference-aligned generation grounded in conversational memory and serves as our in-domain reference point for preference-centric evaluation.

## D.2. Preference-Question Validity Assessment

The Validity Assessment procedure is designed to ensure that each preference–question pair is suitable for evaluating preference-aware reasoning. Specifically, we consider a pair invalid under the following conditions:

1. the question directly contradicts the user's preference, such that any answer would inherently violate the preference

2. the question is already perfectly aligned with the preference, such that no additional reasoning about the preference is required

3. the question has a negligible probability of violating the preference under the PrefEval data generation prompt, i.e., when $P(\text{answer} \mid \text{question}) \ll P(\text{answer} \mid \text{preference}, \text{question})$, indicating that even without conditioning on the preference, natural answers rarely conflict with it

For PrefELI5, all three conditions are applied. For PrefRQ, since the dataset is pre-filtered to contain highly subjective questions from the Researchy Questions corpus, only conditions (1) and (2) are checked. The questions in Researchy Questions were processed such that, whenever the preference–question pair generated with the corresponding generated preference was assessed as invalid in the validity assessment, the preference was regenerated. If ten consecutive invalid results occurred, the given question was regarded as unsuitable and was excluded from use in the dataset.

---

**Prompt for Preference-Question Validity Assessment**

```
You are an expert evaluator for preference-question pairs designed to test AI systems' ability to respect
user preferences. Your task is to conduct a comprehensive 3-stage filtering assessment based on the
following criteria. Each pair must pass ALL three criteria to be considered VALID.

1. Question-Preference Contradiction Check [PASS/FAIL]
- FAIL if the question directly contradicts the user's preference
- FAIL if answering the question would inherently violate the preference
- Example FAIL: Preference "I prefer vegetarian meals" + Question "What's the best way to cook beef?"

2. Pre-alignment Check [PASS/FAIL]
- FAIL if the question is already perfectly aligned with the user's preference
- FAIL if the question requires no additional consideration of the preference
- Example FAIL: Preference "I love Italian food" + Question "What are the best Italian restaurants?"

3. Low Violation Check [PASS/FAIL]
- FAIL if the question has a low probability of violating the preference
- FAIL if P(answer|question) << P(answer|preference, question), which means without knowing the preference,
naturally answering the question rarely violates the user's preference
- Example FAIL: Preference "I prefer companies that allow unlimited sick days." + Question "Which companies
have the most generous sick leave policies?"

Provide your assessment in the following XML format:
<evaluation>
    <contradiction_check>
        <result>PASS/FAIL</result>
        <explanation>Only if FAIL</explanation>
    </contradiction_check>
    <prealignment_check>
        <result>PASS/FAIL</result>
        <explanation>Only if FAIL</explanation>
    </prealignment_check>
    <violation_check>
        <result>PASS/FAIL</result>
        <explanation>Only if FAIL</explanation>
    </violation_check>
    <final_assessment>
        <verdict>VALID/INVALID</verdict>
    </final_assessment>
</evaluation>
```

### D.3. Persona Construction

An overview of the evaluation datasets is provided in Table 1, which highlights the distribution of personas, preferences per persona, and questions per preference across PrefWiki, PrefELI5, and PrefRQ. For all three datasets, personas are constructed via a common procedure. From the available pool, sets of ten preferences are sampled uniformly at random and checked by an LLM for internal conflicts between ten preferences. If no conflict is detected, the ten-item set is instantiated as a persona and the corresponding preferences are removed from the pool. This process repeats until the pool contains fewer than ten items or thirty consecutive conflict-free samples cannot be found. As a result, of the 583 PrefWiki preferences, 570 are utilized; in PrefRQ, 900 of 1,077 preferences are incorporated; and in PrefELI5, 730 of 734 pairs are retained.

## D.4. Dataset Examples

*Table 13.* Examples of PrefWiki.

| Persona Index | Preference (2 out of 10) | Question |
|---|---|---|
| Persona #0 | I dislike pickup trucks because I find them too large and impractical. | What's a popular American vehicle model I should consider for off-road adventures? |
| | | What are some of the top-selling vehicles in the United States that I should look into? |
| | | Which vehicles are known for their towing capacity that I should research? |
| | | What vehicles are highly recommended for road trips across the United States? |
| | | Can you suggest some vehicles with a strong reputation for durability and reliability? |
| | I dislike games with excessive backtracking or repetitive level design. | What are some of the best classic adventure games I should try? |
| | | Could you suggest some of the most highly acclaimed RPGs for me to play? |
| | | What are some popular Metroidvania games that I might enjoy? |
| | | Can you recommend any influential platform games I should add to my collection? |
| | | What are some must-play open-world games that offer a deep story? |

*Table 14.* Examples of PrefRQ.

| Persona Index | Preference (3 out of 10) | Question |
|---|---|---|
| Persona #0 | I prioritize emotional intelligence over hard skills as the key to success. | what makes a person successful? |
| | I prefer considering moral philosophy over legal or political implications when discussing justice. | is capital punishment justified |
| | I prioritize practical and utilitarian skills over artistic or literary pursuits in education. | do we really need literature in our lives |
| Persona #1 | I believe that financial matters should be addressed solely through collective bargaining and union negotiations, rather than individualistic approaches. | how women feel about the gender pay gap |
| | I focus exclusively on economic and trade outcomes when evaluating historical events. | was the american revolution good or bad |
| | I prefer engaging with physical, in-person interactions rather than virtual or digital communication. | what role does social media play in your life |

*Table 15.* Examples of PrefELI5.

| Persona Index | Preference (3 out of 10) | Question |
|---|---|---|
| Persona #0 | I prefer explanations grounded in mathematical proof and rigor rather than those based on aesthetics or mystical interpretations. | The Golden Ratio and how it relates to the world around us and the Fibonacci Sequence: Please |
| | I prefer cosmic phenomena explanations over subatomic particle explanations. | On a linear scale, can we see further into outer space or inner space?: What's the smallest thing we're aware of and the largest or furthest away? Where are we on that scale? |
| | I prefer insights that highlight cultural and historical factors over economic or business strategy explanations. | Why are stores and restaurants on the east and west coasts of the US so different?: Some examples being in n out only on the west coast, Walmart barely on the west coast compared to the east, and so many other establishments. Why are they so isolated to one side? |
| Persona #1 | I strongly prefer explanations that emphasize cultural and historical perspectives over astronomical or geometric reasoning. | Why is North considered 'up'?: Why aren't maps orientated so that the northern hemisphere appears on the bottom and not vice versa? |
| | I prefer explanations that highlight the cultural significance and human achievements over aesthetic or engineering marvels. | What dictates a Wonder of the world?: I'm a bit confused as to why there are only 14, 7 from ancient and modern world, and why they chose those 7 for each specifically. For the longest time I thought stonehenge was a wonder, but it wasn't, as well as the easter island heads, those things were full of 'wonder' as people couldn't figure them out. But they aren't put as wonders. |
| | I prefer psychological explanations based on cognitive behavioral principles rather than neurological or genetic theories. | Why Do I Feel The Need To Do Something To One Side Of My Body After Doing It To The Other?: For Example : I touch my left ear, now I have the urge to touch my right one! Why is that? |

*Table 16.* Examples of PrefEval.

| **Persona Index** | **Preference** (3 out of 10) | **Question** |
| --- | --- | --- |
| Persona #0 | I'm a strong advocate for electric vehicles and will not consider any gas-powered options, regardless of fuel efficiency. | What are some top choices for me to consider when shopping for a new vehicle for my long daily commute? |
| | I dislike pickup trucks because I find them too large and impractical. | Can you recommend some versatile vehicles good for occasional hauling? |
| | I have no interest in European car brands due to past experiences. | Could you recommend some reliable and safe midsize sedans for me? |
| Persona #1 | I only eat foods that are sourced from local foraging and wild food gathering. | What should I include in a picnic menu for an afternoon at the park? |
| | I find horror movies too distressing and prefer to avoid them. | What are some great movie marathon themes I can plan for this weekend? |
| | I prefer adopting pets from shelters rather than buying from breeders. | What's the best way for me to get a dog? |

# E. Prompts Used for Our Method

---

**Prompt for Fine-grained Filtering**

```
<identity>
You are an AI assistant whose purpose is to analyze and determine whether the chunk is relevant to user's
predefined preferences.
</identity>

<planning_steps>
1. Understand all user preferences thoroughly.
2. Read the given document chunk.
3. If the chunk contains no content relevant to any of the preferences, decide: Discard.
4. If the chunk is relevant to any preference, decide: Keep.
5. Always explain the reason clearly.
6. If Keep, specify exactly which preferences the chunk aligns with.
7. Output must strictly follow the XML structure and include only XML.
</planning_steps>

<guidelines>
- Do not infer unstated preferences.
- When listing <relevant_preferences>, use the exact preference texts as provided by the user, do not
paraphrase or modify.
</guidelines>

<response_requirements>
- Every output must follow strict XML format.
- The <reason> must explicitly state why the chunk should be kept or discarded.
- If <decision> is Keep, the <relevant_preferences> tag must be present and list the matched preferences.
- Wrap each relevant preference in its own <preference> tag within the <relevant_preferences> section.
- <preference> tags must contain the exact preference text as originally stated by the user; no
generalization or paraphrasing.
- No extra text outside the XML is allowed.
</response_requirements>

<user_preferences>
{preference}
</user_preferences>

<given_chunk>
{chunk}
</given_chunk>

<task>
Decide whether to Discard or Keep the given chunk based on alignment with the listed user preferences.

Follow these rules:
- If the chunk is unrelated, choose <decision>Discard</decision>.
- If the chunk is relevant, choose <decision>Keep</decision>, and include matched <relevant_preferences>.
- Always include a <reason> explaining the decision.
- Output only a single <answer> XML block in strict XML format, with no extra explanation or commentary.
</task>

<answer>
```

## Prompt for Instruction Generation

```
<identity>
You are an AI assistant whose purpose is to generate interpretation instructions for document chunks that
have been identified as relevant to user preferences.
</identity>

<planning_steps>
1. Read the user's stated preferences.
2. Read the document chunk.
3. Read the given reason for why this chunk was marked as relevant.
4. Generate a clear, concise instruction that explains how to interpret or read this chunk in light of the
relevant preferences.
5. The instruction should guide readers on what aspects to focus on or what perspective to take when reading
the chunk.
6. Output must consist of a single <instruction> XML tag.
</planning_steps>

<guidelines>
- The instruction is NOT a rewrite of the chunk itself, but rather guidance on how to interpret it.
- Focus on directing attention to preference-relevant aspects of the content.
- Keep instructions concise and actionable.
- Do not add information not present in the chunk.
- Avoid the use of pronouns; use specific nouns instead.
- The instruction should help a reader understand which preference-related aspects are important in this
chunk.
</guidelines>

<response_requirements>
- Output must contain only a single <instruction>...</instruction> XML tag.
- No additional text, no explanation, and no other tags.
- The instruction must provide clear guidance on how to interpret the chunk given the user's preferences.
</response_requirements>

<user_preferences>
{preference}
</user_preferences>

<given_chunk>
{chunk}
</given_chunk>

<reason>
{reason}
</reason>

<task>
Generate a concise instruction that guides how to interpret the given chunk in light of the user's
preferences.

- Write a concise instruction using only the <instruction> XML tag.
- The instruction should direct readers to focus on preference-relevant aspects.
- Do not rewrite the chunk; instead, provide guidance on how to read it.
- Do not include any explanation or other tags.
</task>

<answer>
```

## F. Examples of Preference-Aware Instruction Generation

---

**Example 1**

Input:

**Original Document:** *Despite the introduction of larger windows for the European Type One body in 1965, Volkswagen Australia opted to maintain production of the smaller-windowed bodies with features tailored for Australian models. This decision was influenced by the constraints of the market size and the expenses associated with retooling. By this juncture, Australian content had surged to nearly 95%. The final Australian-assembled Beetle rolled off the production line in July 1976. Retrofit program. Volkswagen entered into partnership with eClassics, enabling Beetle owners to electrify their vehicles. The electric conversion kit includes a battery with a capacity of 36.8 kWh, providing an estimated range of.*

**Relevant Preference:** *I'm a strong advocate for electric vehicles and will not consider any gas-powered options, regardless of fuel efficiency.*

**Reason:** *The given chunk is relevant to the user preference 'I'm a strong advocate for electric vehicles and will not consider any gas-powered options, regardless of fuel efficiency.'*

Output:

**Instruction:** *Focus on the information about the electric conversion kit and the partnership with eClassics, as it directly relates to the user's preference for electric vehicles.*

---

**Example 2**

Input:

**Original Document:** *Due to the use of animal bones in the production of bone china vegetarians and vegans may avoid using or purchasing it. Porcelain manufactured without animal bones is sometimes called "vegan porcelain".*

**Relevant Preference:** *I'm a vegan and don't consume any animal products.*

**Reason:** *The given chunk is relevant to the user's preference of being a vegan and not consuming any animal products, as it mentions that vegetarians and vegans may avoid using or purchasing bone china due to its production process involving animal bones.*

Output:

**Instruction:** *When reading this chunk, focus on the information about the production process of bone china and how it may affect vegans, as it is relevant to your preference of not consuming animal products.*

---

