# OpenReview forum: "From Volume to Value: Preference-Aligned Memory Construction for On-Device RAG"
_ICML.cc/2026/Conference — ICML 2026 regular_

### Official Review · Reviewer_3WSS · 2026-03-09

**Soundness:** 2
**Presentation:** 3
**Significance:** 2
**Originality:** 3
**Overall Recommendation:** 4
**Confidence:** 3

**Summary:**

This paper studies a timely and practically important problem: how to make personalized RAG work under tight on-device memory constraints by deciding which documents are worth storing for a given user. The proposed method, EPIC, combines semantic filtering, preference-aware verification, and query steering in a coherent pipeline that aims to preserve user-relevant content while reducing storage and retrieval cost. The empirical results suggest that the approach is effective and consistently improves preference-following performance under constrained budgets.

**Compliance With Llm Reviewing Policy:**

Affirmed.

**Key Questions For Authors:**

Please address the concerns in the weaknesses.

**Limitations:**

yes

**Strengths And Weaknesses:**

Strengths

1. The paper addresses an important and realistic problem setting: personalized RAG under strict on-device memory and efficiency constraints, which is highly relevant for privacy-sensitive and resource-limited applications.

2. The method is well structured and intuitively motivated; the different stages of the pipeline fit together naturally and make the overall approach easy to understand.

3. The experimental evaluation is comprehensive and practically meaningful, covering effectiveness, memory footprint, latency, indexing cost, ablations, and on-device deployment results.

Weaknesses

1. The novelty is moderate: the main contribution is a thoughtful integration of existing ingredients for a new application setting, rather than a sharply new modeling or algorithmic idea.

2. The evaluation depends substantially on synthetic or semi-synthetic preference constructions and LLM-based judging, so the real-world validity of the reported gains is not fully established.

3. Some claims about general on-device applicability feel slightly overstated, since the deployment study is limited to a specific hardware setup and does not yet fully demonstrate robustness across a broader range of consumer devices.

---

> ### Author Rebuttal · Authors · 2026-03-31
>
> Dear Reviewer 3WSS,
>
> Thank you for your thoughtful and constructive review. We sincerely appreciate your recognition of the paper’s practical setting, clear pipeline design, and broad evaluation. Below, we respond to the main concerns you raised.
>
> ---
>
> ### **[W1] Novelty**
>
> Thank you for the comment. We respectfully believe that our contribution is not merely a combination of existing ingredients applied to a new setting. Instead, we identify a **new systems problem** in on-device personalized RAG: the key bottleneck lies not only in retrieval but also in memory construction under strict on-device constraints.
>
> To address this problem, we propose a **coherent system design** in which the **three components tackle distinct but tightly coupled challenges in this new problem setting**: `Sec. 3.1` improves memory efficiency, `Sec. 3.2` improves personalization accuracy, and `Sec. 3.3` improves retrieval/runtime efficiency with minimal additional cost. These components are not simply additive; they are designed to work together around the central challenge of on-device memory construction. As also noted by Reviewer **1AuZ (S2)**, the method is “well-motivated and internally coherent.” We therefore believe that the strong empirical gains come from formulating a new problem and introducing a tailored solution for it, rather than from a straightforward combination of existing techniques.
>
> ---
>
> ### **[W2] Evaluation Validity**
>
> Thank you for the comment. We follow **PrefEval** [r1] in three components of benchmark construction that support its real-world validity: the preference set, the question-preference (Q-P) pair generation pipeline, and the Q-P pair validation procedure. In PrefEval, this construction methodology was supported by its **Extensive Manual Filtering Process** with **human-labeled** examples. As described in `Appendix D`, we adopt the same framework in our benchmark construction. Therefore, our benchmark construction inherits its validity from the PrefEval methodology.
>
> Concretely, as described in Appendix D.1, PrefWiki directly reuses the PrefEval preference set and follows the same Q-P pair generation and validation pipeline. PrefRQ and PrefELI5 also follow the same Q-P generation and validation procedure, but use questions sourced from **Researchy Questions** [r2] and **ELI5** [r3], respectively.
>
> For evaluation, we also follow PrefEval’s LLM-judge protocol, as detailed in `Appendix B.3`. **This protocol was manually validated in PrefEval on 200 randomly sampled evaluations, with a 5% error rate against human judgment**. We use Llama-3.3-70B-Instruct as the judge model, and recent LLM-as-a-judge evidence [r4] suggests that strong 70B-scale judges show reasonable alignment with human judgments. We therefore view this as a practical and reasonably supported protocol for evaluating preference following.
>
> [r1] Do LLMs Recognize Your Preferences? Evaluating Personalized Preference Following in LLMs, Zhao et al., ICLR 2025.
>
> [r2] Researchy Questions: A Dataset of Multi-Perspective, Decompositional Questions for Deep Research, Rosset et al., SIGIR 2025.
>
> [r3] ELI5: Long Form Question Answering, Fan et al., ACL 2019.
>
> [r4] Judging the Judges: Evaluating Alignment and Vulnerabilities in LLMs-as-Judges, Thakur et al., GEM$^2$ 2025.
>
> ---
>
> ### **[W3] On-Device Applicability**
>
> Thank you for the suggestion. The original deployment study aimed to show on-device **feasibility** in a realistic setting, while additional evidence across consumer devices would make EPIC’s practicality more convincing. We therefore conducted additional experiments on consumer devices.
>
> We measured retrieval and indexing latency on a MacBook Pro (M4, 16GB RAM) and a Galaxy Z Flip 6 (Snapdragon 8 Gen 3, 12 GB RAM) using Qwen3-4B-Instruct (4-bit) on PrefWiki. Results are shown below.
>
> |Component|MacBook Pro M4 (ms)|Galaxy Z Flip 6 (ms)|
> |---|---:|---:|
> |*Retrieval Latency*|
> |└ Embedding query|29.40|136.55|
> |└ Preference-Guided Query Steering|0.04|0.22|
> |└ FAISS retrieval|0.07|0.25|
> |**Total Retrieval Latency**|**29.51**|**137.07**|
> ||
> |*Indexing Latency*|
> |└ Embedding items|17.13|443.81|
> |└ Semantic-Based Coarse Filtering|0.01|0.03|
> |└ Preference-Aligned Fine Verification|94.23|621.28|
> |└ Embedding instruction|0.01|0.04|
> |└ Build FAISS|0.00|0.00|
> |**Total Indexing Latency**|**111.39**|**1064.24**|
>
> On MacBook Pro, LLM inference was performed using mlx_lm, with Contriever embeddings accelerated via MPS.
> On Galaxy Z Flip 6, LLM was served by llama.cpp and all components, including LLM inference and Contriever embeddings, operated exclusively on the CPU. Latency could be further reduced with GPU or NPU acceleration.
>
> These results show that EPIC remains lightweight beyond a single deployment platform. With sub-second (or near-second) indexing and millisecond-level retrieval on everyday devices, **EPIC demonstrates that it is not just theoretically sound but practically deployable**. We will include these results in the final version.

---

> > ### Author Rebuttal · Reviewer_3WSS · 2026-04-04
> >
> > Thank you for your detailed rebuttal. After carefully considering your responses, I maintain my original score.

---

> > > ### Author Response · Authors · 2026-04-05
> > >
> > > We sincerely appreciate your positive assessment of our work and your valuable feedback. We are pleased that our rebuttal resolved all your concerns.
> > >
> > > Best regards,
> > >
> > > Authors

---

### Official Review · Reviewer_1AuZ · 2026-03-10

**Soundness:** 3
**Presentation:** 3
**Significance:** 3
**Originality:** 3
**Overall Recommendation:** 4
**Confidence:** 3

**Summary:**

This paper reconceptualizes the principal bottleneck in on-device personalized RAG: rather than asking primarily how to retrieve, it asks what to store. The proposed method, EPIC, consists of a three-stage pipeline:
1. Coarse filtering stage that prunes candidates using preference cues and embedding similarity.
2. LM-based fine verification stage that retains genuinely preference-relevant items and converts them into an instruction-centric memory.
3. Query-steering stage that biases query embeddings toward the preference direction so that retrieval becomes more preference-aligned.

The paper evaluates the approach on four benchmarks (PrefWiki, PrefRQ, PrefELI5, and PrefEval), using preference-following accuracy, memory usage, and retrieval latency as the main metrics, and also reports on-device experiments on a Jetson Orin Nano 8GB. The central claim is that EPIC achieves better preference-following performance and lower retrieval latency than standard RAG and preference-conditioned baselines, while operating under a markedly smaller memory budget.

**Compliance With Llm Reviewing Policy:**

Affirmed.

**Final Justification:**

I have a few reservations, but overall I lean toward weak accept.

**Key Questions For Authors:**

Q1. Did the authors conduct any human evaluation or judge-validation study for the newly introduced benchmarks? The original PrefEval paper helps motivate the importance of this line of evaluation, but since this submission also includes newly generated datasets, separate evidence for judge reliability and dataset validity would be important for establishing soundness.

Q2. How robust is EPIC under implicit, noisy, or conflicting preferences? The current experiments are centered on explicit preference settings, but in realistic on-device agents, preference extraction errors are virtually inevitable. Additional experiments or even a targeted failure analysis in such settings would strengthen the paper’s significance.

**Limitations:**

Yes

**Strengths And Weaknesses:**

### Strengths

S1. The problem formulation is well chosen and practically meaningful. Much of the recent personalization/RAG literature has gravitated toward query rewriting or personalized retrieval, whereas this paper treats memory construction itself as the key lever for personalization. Given that prior work such as PrefEval suggests that preference following is non-trivial even for strong LLMs,  the paper’s *“store less, but store aligned”* framing is both sensible and germane to realistic on-device constraints.

S2. The method is relatively simple, but it is well motivated and internally coherent. The division of function among coarse filtering, LM-based verification/instruction generation, and query steering is clear, and the ablations do a reasonable job of elucidating the contribution of each module. In particular, the coarse filtering stage appears to drive most of the memory reduction, the fine verification stage materially improves accuracy, and the steering component yields an additional gain without increasing storage cost.

S3. As a systems-oriented paper, the empirical case for efficiency is fairly strong. The results in Tables 2–4 and the on-device experiments go beyond showing a marginal accuracy improvement on a server-side setup; they also examine memory footprint, retrieval latency, and streaming updates. That breadth of evaluation is a genuine asset, especially for practical deployment scenarios involving on-device agents.

---
### Weaknesses
W1. My main concern is the evaluation construct. The newly introduced benchmarks (PrefWiki, PrefRQ, and PrefELI5) rely heavily on LLMs during the generation of preferences, questions, and personas, and the final metric is also based on LLM-as-a-judge preference-following accuracy. PrefEval itself is a meaningful benchmark, but it remains somewhat unclear to what extent these newly derived datasets faithfully capture real user logs or messy personal memory as it arises in practice. In other words, while the reported results are promising, the extent to which they generalize to genuine on-device preference memory remains insufficiently substantiated.

W2. The broader narrative of the paper is about a personal agent that infers and exploits preferences from device-generated data, but the empirical study is conducted almost entirely in settings where explicit preferences are already given. As a result, the evidence strongly supports the narrower claim that EPIC can build a compact, preference-aligned index when explicit preferences are available, but it does not yet fully validate the broader personalization story articulated in the introduction. In addition, while the related work discusses memory-side personalization methods (e.g., PEARL-like approaches), the paper does not directly compare against them, which makes me somewhat conservative in assessing the paper’s originality and significance.

W3. (Minor) The description of the comparison setting and data scale is somewhat confusing. In the main text, the paper states that 10,000 documents were sampled from each source due to the memory constraints of the baselines, whereas the appendix describes Wikipedia as using 100,000 articles, yielding 398,861 chunks, and ELI5 as using 20,000 documents, yielding 379,994 chunks. This discrepancy may simply reflect differences in experimental stages or in the definition of corpus units, but as written, it creates ambiguity in interpreting both the scale claims and the efficiency claims.

---

> ### Author Rebuttal · Authors · 2026-03-31
>
> Dear Reviewer 1AuZ,
>
> Thank you for your thoughtful and constructive review. We appreciate your recognition of the paper’s problem formulation, coherent design, and systems-oriented evaluation. Below, we respond to the main concerns and questions you raised.
>
> ---
>
> ### **[W1, Q1] Benchmark Validity and Real-World Relevance**
>
> Thank you for the comment. For the broader benchmark validity and judge reliability, please see our response to Reviewer **3WSS [W2]**.
>
> For real-world relevance, our benchmark suite was designed as a controlled yet diverse testbed for heterogeneous on-device data regimes. Specifically, it covers several practically relevant settings, including **static knowledge sources** (Wikipedia), **noisy web-derived footprints** (Common Crawl), and **conversation histories** (LMSYS-Chat-1M). Together, this benchmark suite allows us to evaluate preference-aware memory construction across heterogeneous sources of personal data commonly arising in on-device settings. We will make this intended scope clearer in the final version so that the benchmark evidence is better aligned with the paper’s claims.
>
> ---
>
> ### **[W2, Q2]  Scope and Robustness to Imperfect Preferences**
>
> Thank you for this thoughtful concern. We agree that the current empirical evidence most directly supports the claim that EPIC can build a compact, preference-aligned index when a preference representation is already available. This is also the intended scope of the paper: as stated in Line 84 (left column), **EPIC assumes that a preference signal is detected by an upstream preference modeling module** ([r1], [r2]). It then studies the complementary problem of **what to store** so that device-resident data remains compact and preference-aligned while still grounding user intent in memory. We will clarify this scope in the introduction and the method overview.
>
> Even within this scope, the available preference signal may still be noisy or internally conflicting in practice. To probe this, we conducted two additional robustness analyses on PrefWiki (averaged over persona IDs 0-9): (1) **noisy preferences**, created with parsing-related errors, character corruption, and random truncation, and evaluated against the original clean preference set; and (2) **conflicting preferences**, created by replacing 5 of the 10 preferences in each persona with their semantic opposites.
>
> |Preference condition|Accuracy (%)|
> |---|---:|
> |Clean pref.|55.2|
> |Noisy pref.|56.2|
> |Conflicting pref.|56.7|
>
> The results show that **performance does not degrade under noisy or conflicting perturbations**, and even improves slightly in this synthetic stress test. A plausible explanation is that noisy preference cues make the coarse filtering stage less restrictive, leaving more candidate chunks available for the later verification and generation. In the conflicting setting, our `Decision Module` also adopts a conservative rule, preserving chunks relevant to **any** of the listed preferences. We do not interpret this as evidence that noisy or conflicting preferences are generally beneficial; rather, it suggests that **EPIC can remain robust when the upstream preference signal is imperfect**.
>
> [r1] Extracting Implicit User Preferences in Conversational Recommender Systems Using Large Language Models, Kim et al., Mathematics 2025.
>
> [r2] Stability-aware Preference Modeling for Sequential Recommendation, Wang et al., ACM Transactions on the Web 2025.
>
> ---
>
> ### **[W2]  Memory-Side Baselines**
>
> Thank you for this important point. We agree that comparing EPIC with memory-side personalization methods would strengthen the paper. However, PEARL is not directly comparable as it assumes user-specific histories and requires training a retriever on personal data, whereas EPIC focuses on what to store under strict on-device memory constraints. We therefore implemented **MemoryBank** [r3] as a more directly comparable memory-based baseline, as it also focuses on long-term personalized memory. We compared it with EPIC on PrefEval using Llama-3.1-8B-Instruct (averaged over persona IDs 0-9):
>
> |Method|Accuracy (%)|Indexing Latency (s)|Retrieval Latency (ms)|Memory (MB)|
> |---|---:|---:|---:|---:|
> |MemoryBank|28.0|4338.76|6.0|36.38|
> |EPIC (ours)|64.0|162.16|3.0|0.41|
>
> These results suggest that, even against a memory-oriented personalization baseline, EPIC achieves substantially stronger preference-following accuracy while also being far more efficient in both memory and latency. We will include this comparison in the final version.
>
> [r3] MemoryBank: Enhancing Large Language Models with Long-Term Memory, Zhong et al., AAAI 2024.
>
> ---
>
> ### **[W3]  Clarification of the Evaluation Scale**
>
> We apologize for the confusion here. The **main-text comparison setting** is the correct one: due to the memory limitations of the baselines, the reported evaluations use 10,000 randomly sampled documents from each source unless stated otherwise. We will fix the number in the final version.

---

> > ### Author Rebuttal · Reviewer_1AuZ · 2026-04-03
> >
> > Thanks for the detailed response. The clarification of scope, the additional robustness analyses, and the comparison against a memory-based baseline significantly strengthen the paper and address several of my concerns. I still consider the external validity of the evaluation only partially resolved, since it remains unclear how faithfully the synthetic benchmark construction and LLM-based evaluation reflect genuine on-device preference memory in practice. Overall, I will maintain the score reflecting the positive assessment of the paper.

---

> > > ### Author Response · Authors · 2026-04-05
> > >
> > > We appreciate your response to our rebuttal. We are glad that the clarification of scope, the additional robustness analyses, and the comparison against a memory-based baseline helped address your concerns and further strengthen the paper.
> > >
> > > Regarding how faithfully the synthetic benchmark construction and LLM-based evaluation reflect genuine on-device preference memory in practice, we would like to clarify two points.
> > >
> > > First, our evaluation is grounded in the **validated preference-following benchmark framework of PrefEval**, which studies preference following through curated preference-query constructions, extensive filtering, and human-checked LLM-based evaluation in a controlled setting. The underlying preference-following validation logic is therefore not something we introduce from scratch, but rather a validated foundation that we adopt for our evaluation.
> > >
> > > Second, our extension is not to the preference-following validation logic itself, but to the practical operating conditions under which it is evaluated. One important dimension is the **heterogeneous nature of incoming data on personal devices**. Rather than evaluating preference following only within a single conversational setting, we extend evaluation to multiple data conditions that users may realistically encounter in on-device environments, including **static knowledge sources**, **noisy web-derived traces**, and **conversational histories**. This broadens the evaluation from a single-session setting to heterogeneous information conditions that are more relevant to practical on-device memory construction.
> > >
> > > Another important dimension is **dynamic usage over time**. To better reflect the evolving nature of on-device preference memory in practice, our **On-Device Experiment** (detailed in `Section 5.4`) evaluates a continual on-device usage scenario in which new documents continuously arrive over time while the active preference profile stochastically changes. This experiment is intended to capture a practically important characteristic of on-device preference memory: memory must be updated continually as user preferences evolve and new information accumulates under device constraints. In the final version, we will make it more explicit that `Section 5.4` serves as a **dynamic evaluation protocol**.
> > >
> > > Taken together, these choices make our evaluation more representative of practical on-device preference memory conditions while remaining grounded in a validated preference-following framework. In this sense, our benchmark should be understood as a controlled proxy rather than a direct replica of real-world on-device interaction traces. At the same time, as also acknowledged in PrefEval, evaluation with raw user logs and real user preferences remains an important next step for capturing richer and more nuanced user interactions. We will make this intended scope and future direction more explicit in the final version.
> > >
> > > Thank you again for the constructive discussion. We will carefully reflect these points in the final version. Please feel free to share any additional thoughts or follow-up questions.
> > >
> > > Best,
> > >
> > > Authors

---

### Official Review · Reviewer_UCpa · 2026-03-11

**Soundness:** 3
**Presentation:** 3
**Significance:** 4
**Originality:** 3
**Overall Recommendation:** 4
**Confidence:** 4

**Summary:**

This paper studies a relatively new and interesting problem: under tight storage budgets, how can we determine which personalized memories are worth storing on-device? To address this question, the authors propose a framework called EPIC, which consists of three main stages: Semantic-Based Coarse Filtering, Preference-Aligned Fine Verification, and Preference-Guided Query Steering. The experimental results show that the proposed method uses less storage space, achieves higher efficiency, and leads to better personalization performance.

**Compliance With Llm Reviewing Policy:**

Affirmed.

**Key Questions For Authors:**

please refer to the weaknesses section

**Limitations:**

no limitation section is included

**Strengths And Weaknesses:**

Strengths

1. The figures are well designed, and the visualizations are clear.
2. The overall organization of the paper is sound.
3. The paper introduces several datasets that are well targeted to the problem.

Weaknesses

1. Around Line 199, it is unclear whether the user preference is assumed to already exist or whether it needs to be estimated from data. This point should be clarified.
2. Based on the results, the component in Section 3.2 seems to contribute the most to the overall improvement. This raises the question of whether other baselines could also achieve noticeable gains if they applied a similar memory filtering step.
3. I did not find experiments on hyperparameter sensitivity. For example, how is the threshold in Equation (1) chosen, and how do different threshold values affect the final performance?
4. The implementation details could be described more thoroughly. This is very important for follow-up research and reproducibility.

---

> ### Author Rebuttal · Authors · 2026-03-31
>
> Dear Reviewer UCpa,
>
> Thank you for your thoughtful and constructive review. We appreciate your positive comments on the clarity and organization of the paper, along with your recognition of the value of the datasets. Below, we respond to the main concerns you raised.
>
> ---
>
> ### **[W1] Clarification of the preference assumption**
>
> We apologize for the ambiguity. In the current paper, EPIC assumes that a user preference representation is already available before indexing. Our goal is to address a complementary but distinct problem: **given an available preference representation, how to construct a compact, preference-aligned on-device memory under tight resource constraints**. Accordingly, EPIC focuses on preference-aligned memory construction and retrieval, rather than on inferring user preferences from raw interaction data within the framework itself.
>
> We agree that this assumption should have been stated more explicitly. In the final version, we will clarify this point at the beginning of `Section 3.1` and more clearly separate the roles of preference extraction and preference-aligned indexing/retrieval, which is the primary focus of EPIC.
>
> ---
>
> ### **[W2] Applicability to Other Baselines**
>
> Thank you for the suggestion. Our additional experiment shows that these components can be applied to another baseline in a plug-and-play manner and serve distinct roles: **Semantic-Based Coarse Filtering** (`Sec. 3.1`; **C**) mainly improves memory and latency efficiency, **Preference-Aligned Fine Verification** (`Sec. 3.2`; **F**) helps preserve preference relevance and overall accuracy, and **Preference-Guided Query Steering** (`Sec. 3.3`; **S**) provides an additional retrieval-time gain without increasing storage cost.
>
> To test this directly, we applied these components to **HippoRAG2** on PrefWiki under the **F**, **CF**, and **CFS** settings using Llama-3.1-8B-Instruct. The results below are averaged over 10 profiles (persona IDs 0-9).
>
> |Method|Accuracy (%)|Indexing Latency (s)|Retrieval Latency (ms)|Memory (MB)|
> |---|---:|---:|---:|---:|
> |HippoRAG2|42.9|4726.25|615.00|2896.50|
> |HippoRAG2 + **F**|56.8|2215.60|523.27|22.79|
> |HippoRAG2 + **CF**|52.2|**302.99**|444.94|**3.74**|
> |HippoRAG2 + **CFS**|**60.0**|**302.99**|**439.80**|**3.74**|
>
> These results further support the complementary roles of the three components: **F** improves accuracy (**42.9%→56.8%**), **C** delivers the main efficiency gain (**22.79 MB→3.74 MB**; **2215.60 s→302.99 s**), and **S** further improves accuracy (**52.2%→60.0%**) without increasing memory or indexing cost. We will include this analysis in the final version.
>
> ---
>
> ### **[W3] Threshold Sensitivity and Selection**
>
> Thank you for the question. To examine threshold sensitivity directly, we conducted additional analysis over `τ ∈ {0.0, 0.1, 0.2, 0.3, 0.4}` on all four datasets using Llama-3.1-8B-Instruct. The results below are averaged over 10 profiles (persona IDs 0-9) for each dataset.
>
> |Dataset|Metric|0.0|0.1|0.2|**0.3**|0.4|
> |:---|:---|:---|:---|:---|:---|:---|
> |**PrefWiki**|Acc. (%)|45.0|45.0|68.8|**55.2**|47.2|
> ||Lat. (s)|3600.12|3600.12|1847.11|**157.76**|3.20|
> ||Mem. (MB)|1.61|1.61|1.89|**0.27**|0.01|
> |**PrefEval**|Acc. (%)|72.0|72.0|77.0|**64.0**|37.0|
> ||Lat. (s)|1269.30|1269.30|944.00|**162.16**|12.76|
> ||Mem. (MB)|2.47|2.47|1.78|**0.37**|0.02|
> |**PrefELI5**|Acc. (%)|86.0|86.0|87.0|**88.0**|84.0|
> ||Lat. (s)|3986.07|3986.07|3394.89|**605.73**|31.01|
> ||Mem. (MB)|13.85|13.85|12.27|**4.50**|0.24|
> |**PrefRQ**|Acc. (%)|86.0|86.0|84.0|**83.0**|71.11|
> ||Lat. (s)|3783.50|3783.50|2106.68|**81.89**|5.93|
> ||Mem. (MB)|6.29|6.29|6.12|**0.92**|0.02|
>
> The results show a clear trade-off: smaller thresholds retain too many memories and lead to very high latency, whereas larger thresholds prune more aggressively and can hurt accuracy. Since EPIC is designed for **resource-constrained deployment**, we select `0.3` as the default operating point because it provides the most practical balance among accuracy, latency, and memory, although the preferred threshold may vary depending on the user’s budget and deployment goal. We will include this sensitivity analysis in the final version.
>
> ---
>
> ### **[W4] Reproducibility and Implementation Details**
>
> Thank you for the suggestion. We agree that the current draft could be strengthened by including further implementation details.
>
> While key details are already provided in `Appendix C`, including corpus/chunking details and the memory/latency measurement protocol, we will make this more explicit in the main text. To further improve reproducibility, we will add clearer step-by-step algorithmic descriptions (including pseudocode) for indexing and retrieval, a more explicit explanation of the threshold selection procedure, and clearer documentation of the newly added ablations and baseline settings.
>
> We will revise the paper accordingly and would be happy to include any additional implementation details you find especially useful.

---

> > ### Author Rebuttal · Reviewer_UCpa · 2026-04-01
> >
> > All my previous concerns are resolved. Thanks the authors for the rebuttal, i will keep my positive score.

---

> > > ### Author Response · Authors · 2026-04-05
> > >
> > > We sincerely appreciate your positive assessment of our work and your valuable feedback. We are pleased that our rebuttal resolved all your concerns.
> > >
> > > Best regards,
> > >
> > > Authors

---

### Official Review · Reviewer_2Z3F · 2026-03-17

**Soundness:** 3
**Presentation:** 2
**Significance:** 3
**Originality:** 3
**Overall Recommendation:** 4
**Confidence:** 3

**Summary:**

This paper introduces EPIC, a novel preference-aligned memory construction method designed specifically for on-device Retrieval-Augmented Generation (RAG). By integrating preference learning directly into the index construction phase rather than merely post-retrieval generation, the authors successfully filter out non-essential data. The empirical results are highly impressive, achieving up to a 2404x reduction in memory footprint and significant latency improvements while simultaneously boosting preference-following accuracy on four benchmark datasets.


# 4. Questions for Rebuttal
1. How does EPIC handle "out-of-distribution" user queries where the required knowledge was filtered out during the preference-aligned index construction?
2. Could the authors provide an ablation study or discussion on the trade-off between the aggressiveness of preference filtering and the recall drop on rare/factual edge cases?

**Compliance With Llm Reviewing Policy:**

Affirmed.

**Key Questions For Authors:**

1. How does EPIC handle "out-of-distribution" user queries where the required knowledge was filtered out during the preference-aligned index construction?
2. Could the authors provide an ablation study or discussion on the trade-off between the aggressiveness of preference filtering and the recall drop on rare/factual edge cases?

**Limitations:**

yes

**Strengths And Weaknesses:**

Strength
- **High Practical Relevance:** The challenge of memory constraints in on-device RAG is a pressing issue in deploying LLMs to edge devices. This paper directly addresses a significant bottleneck.
- **Innovative Methodology:** Shifting the preference alignment from the generation phase to the memory construction/indexing phase is a clever and effective paradigm shift.
- **Strong Empirical Results:** The reported 2404x memory reduction with sub-megabyte footprint on edge devices is a remarkable engineering and algorithmic achievement.

Weakness
- **Risk of Information Loss:** By aggressively filtering memory based on specific preference alignments, the system might overfit to seen preferences, potentially failing to retrieve factually correct but "non-preferred" tail knowledge.
- **Baseline Comparisons:** While the memory reduction is massive, the baselines could be strengthened by including the most recent ultra-high-compression vector quantization (e.g., advanced PQ or binary embeddings) methods for a fairer comparison in constrained environments.

---

> ### Author Rebuttal · Authors · 2026-03-31
>
> Dear Reviewer 2Z3F,
>
> Thank you for your thoughtful and constructive review. We appreciate your recognition of the paper’s practical relevance, the shift from generation-time alignment to memory construction, and the paper’s strong empirical gains. Below, we respond to the main concerns and questions you raised.
>
> ---
>
> ### **[W1, Q1, Q2] Coverage Trade-off and Out-of-Distribution Queries**
>
> Thank you for raising this important point. Under a strict memory budget, there is an inherent trade-off between preference alignment and knowledge coverage: more aggressive filtering can remove factually useful but less preference-aligned tail knowledge.
>
> EPIC is not intended to preserve all possible knowledge under a fixed budget. Rather, as noted in Line 62 (left column), **it prioritizes what to store so that the resulting memory remains compact and preference-aligned for on-device deployment**. In this sense, EPIC addresses a constrained retention problem, and such a design can inevitably introduce information loss on out-of-distribution or weakly preference-matched queries. To investigate this trade-off, we have additionally conducted filtering-threshold analysis (see our response to Reviewer **UCpa [W3]**): our most factual-heavy setting, accuracy drops from **68.8%** to **55.2%** to **47.2%** as the threshold increases from `0.2` to `0.3` to `0.4`, while latency and memory reduce sharply. Although our current benchmarks do not explicitly target rare or tail factual cases, this result suggests that overly aggressive filtering may remove chunks that are not strongly preference-matched but are still needed to support a factually correct answer.
>
> In a practical deployment, however, EPIC can be paired with a **query-time fallback path**, where low query-preference similarity is used to detect weakly matched queries and route them to a more general pipeline (e.g., cloud-based general search engines). A complementary option is a **multi-tiered memory design** with an ultra-compact, strictly preference-aligned tier for personalized responses and a less aggressively filtered tier for broader factual coverage. We view these multi-tiered hybrid designs as a natural extension beyond the present scope. We will add this discussion to the final version.
>
> ---
>
> ### **[W2]  Baseline Comparisons with High-Compression Retrieval Methods**
>
> Thank you for the suggestion. In the main experiments, we initially did not include ultra-high-compression methods that focus on compressing the vector index, as they are fundamentally on a different design axis from EPIC. While EPIC minimizes the memory footprint by selectively determining what to store, vector quantization methods compress the numerical representation of the embeddings themselves.
>
> Still, we agree that it’s beneficial to understand the performance of such methods in our setup. We additionally implemented three representative baselines: **BPR** [r1] for hashing/binary-code retrieval, **RaBitQ** [r2] for vector quantization, and **TurboQuant** [r3] for online vector quantization. We compared them against EPIC on PrefWiki using Llama-3.1-8B-Instruct. The results below are averaged over 10 profiles (persona IDs 0-9).
>
> |Method|Accuracy (%)|Indexing Latency (s)|Retrieval Latency (ms)|Memory (MB)|
> |---|---:|---:|---:|---:|
> |BPR|41.2|88.85|23.31|28.86 (4.93×)|
> |RaBitQ|40.8|91.34|**1.02**|30.06 (4.73×)|
> |TurboQuant 2-bit|44.0|22.63|547.82|34.91 (4.07×)|
> |TurboQuant 4-bit|43.8|**13.80**|548.03|42.21 (3.37×)|
> |EPIC (ours)|**55.2**|246.38|3.00|**0.27** (526.52×)|
>
> Relative to the Contriever baseline memory footprint (142.16 MB = 116.95 MB index + 25.21 MB retained chunks), BPR, RaBitQ, TurboQuant 2-bit, and 4-bit achieve 4.93×, 4.73×, 4.07×, and 3.37× compression, respectively, while EPIC reaches 526.52× compression.
>
> This highlights an important distinction: BPR, RaBitQ, and TurboQuant **mainly compress the vector index itself** (from 116.95 MB to 3.65 MB, 4.85 MB, 9.7 MB, and 17.0 MB, respectively), while the retained chunk storage remains unchanged at 25.21 MB, resulting in 28.86-42.21 MB total memory. By contrast, **EPIC reduces the memory footprint by changing what is retained in memory in the first place**, which allows it to reduce both the stored memory set and the retrieval burden. We will include these additional baselines and discussion in the final version to better illustrate EPIC’s unique advantages in addressing the fundamental 'what-to-store' problem under strict on-device budget constraints.
>
> [r1] Efficient Passage Retrieval with Hashing for Open-Domain Question Answering, Yamada et al., ACL-IJCNLP 2021.
>
> [r2] RaBitQ: Quantizing High-Dimensional Vectors with a Theoretical Error Bound for Approximate Nearest Neighbor Search, Gao et al., Proc. ACM Manag. Data 2024.
>
> [r3] TurboQuant: Online Vector Quantization with Near-optimal Distortion Rate, Zandieh et al., ICLR 2026.

---

> > ### Author Rebuttal · Reviewer_2Z3F · 2026-04-06
> >
> > Thank you for your detailed answer. After carefully considering your responses, I decided to maintain my original score.

---

> > > ### Author Response · Authors · 2026-04-06
> > >
> > > We sincerely appreciate your positive assessment of our work and your valuable feedback. We are pleased that our rebuttal resolved all your concerns.
> > >
> > > Best regards,
> > >
> > > Authors

---

### Decision · Program_Chairs · 2026-04-30

**Decision:**

Accept (regular)

**Comment:**

The paper has some merits in terms of its high practical relevance, innovative paradigm shift, and strong empirical results. Evaluation is thorough (memory, latency, streaming). For weaknesses—LLM-generated preferences instead of real user logs, explicit preferences rather than inferred, missing comparisons to advanced compression—are addressable in revision and do not undermine the core contribution.